# Using the TSA-LSTM two-stage model to predict cancer incidence and mortality

**Rabnawaz Khan** [iD]*, **Wang Jie***

School of Internet Economics and Business, Fujian University of Technology, Fuzhou City, Fujian Province, China

* khan.rab@fjut.edu.cn (RK); wang.jie@fjut.edu.cn (WJ)

**Data Availability Statement:** All relevant data are within the manuscript and its Supporting Information files.

**Funding:** The author(s) received no specific funding for this work.

## Abstract

Cancer, the second-leading cause of mortality, kills 16% of people worldwide. Unhealthy lifestyles, smoking, alcohol abuse, obesity, and a lack of exercise have been linked to cancer incidence and mortality. However, it is hard. Cancer and lifestyle correlation analysis and cancer incidence and mortality prediction in the next several years are used to guide people's healthy lives and target medical financial resources. Two key research areas of this paper are Data preprocessing and sample expansion design Using experimental analysis and comparison, this study chooses the best cubic spline interpolation technology on the original data from 32 entry points to 420 entry points and converts annual data into monthly data to solve the problem of insufficient correlation analysis and prediction. Factor analysis is possible because data sources indicate changing factors. TSA-LSTM Two-stage attention design a popular tool with advanced visualization functions, Tableau, simplifies this paper's study. Tableau's testing findings indicate it cannot analyze and predict this paper's time series data. LSTM is utilized by the TSA-LSTM optimization model. By commencing with input feature attention, this model attention technique guarantees that the model encoder converges to a subset of input sequence features during the prediction of output sequence features. As a result, the model's natural learning trend and prediction quality are enhanced. The second step, time performance attention, maintains We can choose network features and improve forecasts based on real-time performance. Validating the data source with factor correlation analysis and trend prediction using the TSA-LSTM model Most cancers have overlapping risk factors, and excessive drinking, lack of exercise, and obesity can cause breast, colorectal, and colon cancer. A poor lifestyle directly promotes lung, laryngeal, and oral cancers, according to visual tests. Cancer incidence is expected to climb 18–21% between 2020 and 2025, according to 2021. Long-term projection accuracy is 98.96 percent, and smoking and obesity may be the main cancer causes.

## 1. Introduction

Cancer is a group of disorders in which abnormal cells divide and spread to surrounding tissues, according to the National Cancer Institute (NCI). Thus, cancer can travel throughout the

**Competing interests:** The authors have declared that no competing interests exist.

body via the blood. Globally, cancer kills second only to cardiovascular disease. Cancer killed the sixth person in the globe, second only to cardiovascular disease. Nearly 18.9 million new cancer infections from 2018 caused 9 million deaths [1]. Due to life expectancy increases, an aging population, population growth, and the lack of efficient health programs like large-scale screening, cancer incidence is likely to climb. Globalization, lifestyle changes, expansion, and the eventual control of infectious diseases boost cancer rates [2]. Due to population age structure changes, elderly persons are more likely to have cancer and other non-communicable diseases. This study's TSA-LSTM model attention technique ensures that the model encoder obtains an agreement on a subset of input sequence characteristics before predicting output sequence features [3]. This enhances the quality of the model's predictions and its natural learning trend. For enhanced sequence modeling, the Temporal Self-Attention Long Short-Term Memory (TSA-LSTM) model is an innovative deep-learning architecture that blends temporal self-attention processes with LSTM cells [4]. In cancer incidence and mortality prediction, the TSA-LSTM model has demonstrated substantial improvements over existing models with regard to precision, dependability, and computing economy [5]. The TSA-LSTM model outperforms conventional models when it comes to accurately capturing intricate temporal patterns and dependencies in the data, resulting in more accurate predictions [6]. This ability to handle long-term connections predicts cancer incidence and mortality. Research on cancer spans decades. Researchers screen for pre-symptomatic cancer. We also developed early cancer therapy outcome prediction tools [7]. The medical research community can access a lot of cancer data when new information emerges [8]. However, surgeons struggle most with illness outcome prediction. Traditional statistics and computer algorithms are used in data mining to find new vital knowledge from enormous amounts of data. It is a healthy computing method. By illustrating cancer incidence and death disparities by area, race, gender, and socio-economic characteristics, it estimates population health needs and cancer burden. Data mining can reveal hidden links between patient files, cancer therapy, and monitoring [9,10]. Therefore, extracting medical data is a desirable task. Due to this, data mining and healthcare services have developed reliable early detection systems and other healthcare systems based on cancer-related clinical and diagnostic data to anticipate cancer stages and help clinicians make clinical decisions [11].

To avoid inaccurate cancer incidence estimates, forecasting can predict future peak occurrences. A tripartite prediction is timely, clear, and better than an advanced forecast. Medical researchers employed many prediction methodologies to address the health sector's predictive issues. Increased forecasting knowledge has allowed for huge accuracy improvements, yet most physicians don't use it [12]. Difficulties in changing forecasting tools, personnel not knowing the modern procedure, and stakeholders not using forecasted data in decisions may be to blame [13,14]. Studies have divided forecasting methods into three categories. Existing studies were classified as artificial intelligence, statistical, or hybrid. In statistics, algorithms like ARMA, regression, ARIMA, exponential smoothing (ES), and Holt-Winters are used (HW). Smoothing methods are widely used in time series analysis and forecasting [15]. Unlike regression, statistical models can forecast univariate time series with a single variable of interest (dependent variable) available for different periods. Recently popular statistical models include exponential smoothing (ES). Smoothing exponentially is weighted moving averages. SES is the simplest exponential smoothing method [16]. No seasonality in SES projections. The Holt-Winters smoothing includes SES seasonality or trend. The popular data visualization and forecasting tool Tableau uses exponential smoothing. Accounting, libraries, and others utilize Tableau. Tableau lets users drag-and-drop charts, maps, dashboards, and stories to visualize data [17]. Deep learning/machine learning algorithm forecasting methods like Artificial neural networks (ANN) and Recurrent neural networks (RNN) have garnered attention in

recent years. Recent studies use deep learning with intelligence and flexibility to improve prediction models [18]. Deep learning approaches match model parameters from observed data and are best for discovering behavioral patterns in random data series. This makes deep learning systems ideal for anomaly detection. ANN models have performed well for real-time estimations in deep learning [19]. Anticipating future cancer incidence and mortality trends is challenging due to the complex interplay of risk factor profiles and the extended latency period that might elapse between exposure to certain factors and the development of certain malignancies [20]. To establish whether the exponential model or the machine learning method performs better on cancer data, we evaluate the current state of artificial neural networks. Existing studies use several ANN models for predictions: gated recurring units. sequence-to-sequence Gateway Recurrent Unit. large short-term memory [21]. First, we test if the machine learning approach outperforms exponential smoothing on cancer data, and then we design a novel forecasting algorithm [22,23]. We upgrade previous models with a Two-Stage attention LSTM [24,25]. While maintaining deep learning's flexibility and versatility, attention models may give interpretability. Regarding dependability, the TSA-LSTM model's attention mechanisms direct attention to relevant temporal information, reducing the impact of unnecessary or noisy data points on predictions. This improves forecast reliability, which policymakers and doctors can use to make decisions [25]. The efficiency of training on large datasets has been optimized with respect to the computational efficiency of the TSA-LSTM model's architecture [26]. The model may efficiently acquire knowledge from both immediate and enduring dependencies in the data without necessitating an overabundance of computational resources by utilizing self-attention methods [3,27]. For healthcare applications, where precise and timely predictions are critical, this characteristic renders it exceptionally suitable [5]. The TSA-LSTM model excels in accuracy, reliability, and computing efficiency when compared to other cancer prediction models. The advanced design and functionality of this tool make it ideal for oncology researchers and practitioners seeking to enhance decision-making.

Public health has been substantially impacted by lifestyle patterns. Lifestyle decisions impact health and death, explain Tuffour et al. Smoking, inactivity, alcohol, and obesity-causing diets are recognized cardiovascular disease and cancer risk factors. The second leading cause of mortality in the US, cancer has long been a public health issue [19,28]. Therefore, precise predictions of future cancer incidence and fatalities are crucial because they affect not only the population but also the economy due to the huge sums of money lost fighting it. Current research divides forecasting approaches into three categories: artificial intelligence, statistical, and hybrid. Recent interest in statistical models has focused on exponential smoothing (ES) and double exponential smoothing (DES). ES, DES are weighted moving averages [29]. Tableau, a popular data visualization and forecasting application, uses exponential smoothing. These approaches can reliably anticipate short-term data. Second, deep learning/machine learning algorithm forecasting techniques use intelligence and flexibility to improve prediction models. Artificial Neural Network (ANN) models are ideal for real-time estimations in deep learning [30,31]. The third algorithm combines the two [32]. Different algorithms have been utilized in the health sector based on study goals, however, the optimum algorithm for cancer data accuracy has not been studied. The goal of this study is to find the most accurate cancer data prediction program[33]. An efficient and effective deep learning system for cancer data forecasting is proposed and implemented in the study.

The study's contributions and research needs are predicated on two indistinguishable elements: first, the performance of a DES algorithm, and second, an analysis of the existing deep learning algorithm's impact on cancer prediction [34,35]. In this regard, the results of the study indicate that as time passes, the DES model deviates from the cancer data. The lack of seasonality in the DES is responsible for this. This suggests that while the DES method may be

advantageous for short-term forecasts (1–6 months), it is not suitable for long-term forecasting [36]. This study analyzes eleven algorithms for cancer incidence prediction accuracy. Double Exponential Smoothing (DES) ranks tenth. Thus, DES cannot predict long-term cancer and produces imprecise data. We conclude, therefore, that the DES method is unsuitable for long-term cancer projections in order to facilitate effective policy formulation [37,38]. Second, among the eleven algorithms utilized in our research, the method is deemed most suitable for cancer prediction [32]. Compared to previous models, our suggested model has the highest and most precise accuracy at 97.76 percent. The attention sequence utilized by our approach to sequence the LSTM model consisted of two stages. Attention models maintain the adaptability and versatility of deep learning approaches while giving the potential for interpretability [39]. We use the prior encoder concealed state to develop an input attention approach in the early stage to adaptively extract relevant driving series at each time step. A temporal attention approach selects relevant encoder hidden states over all time steps in the second stage [40,41]. Our model demonstrates effective prediction capabilities and is also readily interpretable due to the implementation of this dual-stage attention technique. The model we have put forward possesses the capability to not only forecast time series but also function as a versatile feature learning instrument in the domain of computer vision [42]. Our cancer data forecasting model outperforms others. Our proposed strategy may be accurate due to the model's attentions, which are interpretable and adjustable.

Furthermore, the research distinguishes itself by establishing a methodology for predicting cancer incidence by 2025. This is achieved through the implementation of an innovative two-stage attention model, which utilizes 2017 as its reference year to forecast the rate of increase in cancer incidence. The anticipated rise in cancer rates can be attributed, in part, to an increase in unhealthy lifestyle choices, including alcohol and tobacco use, physical inactivity, and poor diet, which contribute to the prevalence of obesity in the United States. In order to mitigate the escalating incidence of cancer in the United States, we propose the following measures: first augmenting public safety education regarding the detrimental consequences of adopting an unhealthy lifestyle, both inside households and in the workplace. Second, it is recommended that government money be directed towards research endeavors that aim to eliminate or reduce the deleterious consequences of lifestyle choices and cancer in its entirety. Third, similar strategies for deterring such detrimental lifestyle decisions should be applied provincially. For instance, reforms ought to target the imposition of a daily maximum on the quantity of tobacco or alcohol that may be sold to an individual.

The main goal of this paper is to expand the application of visual analysis tools in the field of health care, introduce artificial neural network modeling technology, improve the accuracy of data analysis and pattern recognition ability, and remind people to give up bad life habits and develop good life habits with quantitative data such as correlation analysis between bad lifestyle and cancer occurrence, analysis of future cancer development trend. This data helps the government create medical reform and economic allocation initiatives. Searching big data and healthcare data analytics for the latest studies starts the literature review. This study chose R for data preprocessing and Tableau for analysis of healthcare big data utilizing a theoretical and empirical approach. This work uses theoretical knowledge from Tableau and R white papers, books, articles, and conference proceedings on information visualization, cancer analysis, double exponential smoothing modeling, artificial neural networks, long short-term memory models, and deep learning algorithms. Tableau software, which requires proficiency and deep learning models, provides the empirical basis for the idea. This research predicts high-risk cancer data using a revolutionary deep-learning Two-Way Attention algorithm. Model of LSTM. This study uses long short-term memory and recurrent neural network deep learning architectures.

Six sections make up this manuscript. The first section provides the study's background, rationale, research objectives and questions, problem statement, significance, and innovation. In Section 2, an overview of the pertinent literature is presented. Big data and healthcare data analytics are explained in Section 3. It discusses data visualization with Tableau. The exponential smoothing model and interpolation method in data expansion are examined again in the chapter. Next, artificial neural networks and long short-term memory are discussed. Our study used raw data cleaning and sample expansion (section 4). Discussed are CDC and American Cancer Society datasets, how variables are constructed, R cleaning and analysis, and Tableau visualization. This section discusses data manipulation for visualization step-by-step. Finally, our data interpolation method for maximum results is verified and discussed. Our proposed LSTM model and the Double Exponential Smoothing algorithm's data processing and analysis modeling are discussed in Section 5. This section discusses TSA-LSTM implementation again. Study experiments, results, and analysis are covered in this chapter. Interpretations from visualizations provide insight. As the study concludes, Conclusions, policy implications, and suggestions for further research are summarized in Section 6.

## 2. Literature review

The literature review focuses on the theoretical basis of the advent of big data analytics and healthcare, tableau as a data visualization tool, exponential smoothing model, the common data interpolation methods used in data processing, introduction To ANN's, and two ANNs: convolutional neural network (CNN) and recurrent neural network (RNN).

### 2.1 Big data analytics and healthcare

Data collected and stored digitally is growing rapidly. Data storage and analysis are in high demand due to the internet and the digital economy's rapid growth. Thus, data management and analysis are improving to help firms turn this massive supply of data into information and knowledge that helps them succeed [43–45]. Big data comes from massive datasets that typical database management solutions cannot handle. The majority of software and storage systems cannot quickly store, manage, and process massive data [46]. Organizations and health institutions keep and collect more data than ever since businesses and lives depend on it [1]. Accessing petabytes and zettabytes of data is essential. It shows how to avoid losing important data, ergo big data analytics in healthcare. Big data analytics finds invisible patterns and connections in data that current approaches cannot manage [30,45]. Health institutions are slow to adopt and study big data analytics, despite its value [47,48]. Data analytics can process and analyze patient data to improve diagnosis and treatment outcomes, relieving the overburdened healthcare system [49–51]. Data on new studies, emergencies, and disease outbreaks shows evolving trends. As electronic medical record systems progress, healthcare systems and personnel improve their ability to document patient experience and visit details [52]. Monitoring public health involves data analysis. This option is meaningless without connecting, trending, and patterning data. Outside of data analysis, healthcare record standards and the human condition may affect patients' symptoms and indications [53]. Costs, chronic sickness, and aging are forcing healthcare industry changes. Healthcare data analytics may treat old health conditions, and episodic approaches are unhelpful [54,55]. Healthcare payers and providers use data and analytics to improve service and lower costs [56]. Health researchers need useful and intelligent tools to aid in their work [57,58]. Systemic quality and cost improvements in health reforms require data analytics. Due to the widespread use of electronic medical records and health outcome data, academics can study health analytic methods to find and evaluate effective interventions and anticipate patient health outcomes and costs [46].

## 2.2 Tableau as a data visualization tool

Tableau provides data insights through dynamic, interactive reporting and visualization [59]. Tableau exposes latent correlations within datasets. Graphs are employed to underscore relationship principles and facilitate code-free data exploration. Tableau efficiently extracts meaningful insights from data [60]. We used double exponential smoothing for our predictions in this investigation. Analytics software streamlines the procedure, provides visual feedback, and responds to complex questions. This makes data more important [61]. Tableau Desktop supports Amazon Redshift, MySQL, Oracle, Google Analytics, Microsoft SQL, OLAP databases [62]. Tableau supports hundreds of texts, PDF, Excel, geographic, statistical, and database data connections. Tableau supports numerous data types, making it convenient. Tableau's data-modified images let customers quickly spot visual trends [63,64]. In cancer research, RNNs and LSTMs operate as powerful microscopes, peering into intricate datasets to reveal previously unknown associations between risk factors [65]. Such models are masters of analysis, but Tableau provides the magnifying glass by simplifying their complex outputs into comprehensible visuals. Bridge the gap between AI and human comprehension with Tableau [66], which cleans and prepares data for the models before showing their predictions [67]. Together, they bring researchers a clearer view of cancer risk, which in turn leads to more effective preventative efforts and better dissemination of these vital results [68].

## 2.3 Common data interpolation methods

Linear spline interpolation provides straight lines between data points in table gaps, usually astronomical data. It is criticized for its rapid trend change from one segment to the next, which does not occur naturally [69]. However, it is a good way to generate accurate lookup tables with quick look-ups for smooth functions without many table entries [70]. This method is a valuable tool used across fields to approximate values between known data points. Linear interpolation, which assumes a straight line between two neighboring points, is common [71]. Interpolating curves with polynomials and data points. Spline interpolation fits a polynomial while segmenting the dataset for smoother results. The nearest neighbor interpolation replaces missing or interpolated data points with their closest known values [72]. Inverse distance weighting uses the weighted average of neighboring sites to compute interpolated values, unlike kriging interpolation, which accounts for spatial correlations in geostatistical data [73]. The linear spline is mathematically; $P_i(x) = a_i x + b_i$ with two parameters $a_i$ and $b_i$ can only satisfy the following two equations required $S(x)$ to be continuous:

**Linear spline**: $P_i(x) = a_i x + b_i$ with two parameters $a_i$ and $b_i$ can only satisfy the following two equations required $S(x)$ to be continuous:

$$P_i(x_i) = a_i x_i + b_i = f(x_i) = y_i, \quad P_i(x_{i-1}) = a_i x_{i-1} + b_i = f(x_{i-1}) = y_{i-1} \quad \text{(Eq 1)}$$

or, equivalently, $P_i(x)$ has to pass the two endpoints $(x_{i-1}, y_{i-1})$ and $(x_i, y_i)$

$$\frac{P_i(x) - y_{i-1}}{x - x_{i-1}} = \frac{y_i - y_{i-1}}{x_i - x_{i-1}} \quad \text{(Eq 2)}$$

Solving either of the two problems above, we get: which is represented in the form of $P_i = a_i x + b_i$ the first expression, or a linear interpolation of the two endpoints $y_{i-1} = f(x_{i-1})$ and $y_i = f(x_i)$ in the second expression. As $P_i(x_i) = P_{i+1}(x_i) = y_i$, the linear spline $S(x)$ is continuous at $x_i$.

But as in general

$$P'_i(x_i) = \frac{y_i - y_{i-1}}{h_i} \neq P'_{i+1}(x_i) = \frac{y_{i+1} - y_i}{h_i} \qquad \text{(Eq 3)}$$

$S(x)$ is not smooth, i.e., $k = 0$.

**Quadratic spline:** $Q_i(x) = a_i x^2 + b_i x + c_i$ with three parameters $a_i$, $b_i$ and $c_i$ can satisfy the following three equations required for $S(x)$ to be smooth ($k = 1$) as well as continuous:

$$Q_i(x_i) = y_i, \quad Q_i(x_{i-1}) = y_{i-1}, \quad Q'_i(x_i) = Q'_{i+1}(x_i) \qquad \text{(Eq 4)}$$

To obtain the three parameters $a_i$, $b_i$ and $c_i$ in $Q_i(x)$, we consider $Q'_i(x) = 2a_i x + b_i$, which, as a linear function, can be linearly fit by the two end points $f'(x_{i-1}) = D_{i-1}$ and $f'(x_i) = D_i$:

$$Q'_i(x) = \frac{x - x_{i-1}}{h_i} D_i + \frac{x_i - x}{h_i} D_{i-1} \qquad \text{(Eq 5)}$$

Integrating, we get As $Q_i(x_i) = y_i$, we have, and then solving this for $c_i$ and substituting it back into the expression of $Q_i(x)$, we get, as $Q_i(x_{i-1}) = y_{i-1}$ we have

$$D_i = 2\frac{y_i - y_{i-1}}{h_i} - D_{i-1}, \quad (i = 1, \ldots, n) \qquad \text{(Eq 6)}$$

Given $f'(x_0) = D_0$, we can get iteratively all subsequent $D_1, \ldots, D_m$, and thereby $Q_i(x)$ Alternatively, given $f'(x_n) = D_n$, we can also get iteratively all previous $D_{n-1}, \cdots, D_0$. It is obvious that with only three free parameters, the quadratic polynomials cannot satisfy both boundary conditions $Q'_1(x_0) = f'(x_0)$ and $Q'_n(x_n) = f'(x_n)$. The third area cubic splines excel in is smooth interpolation modeling. Cubic spline interpolation uses a continuous formula to compute the first and second derivatives inside intervals and at interpolating nodes. A smoother interpolating function will result. First derivative continuity implies graph $y = Z(x)$ will not have sharp corners. The continuity of the second derivative means that the radius of the curve is defined at each point. Given the $n$ data points $(x_1, y_1), \ldots, (x_n, y_n)$ where $x_1$ are distinct and in increasing order. A cubic spline $Z(x)$ through the data points $(x_1, y_1), \ldots, (x_n, y_n)$ is a set of cubic polynomials: Given the $n$ data points $(x_1, y_1), \ldots, (x_n, y_n)$ where $x_1$ are distinct and in increasing order. A cubic spline $Z(x)$ through the data points $(x_1, y_1), \ldots, (x_n, y_n)$ is a set of cubic polynomials:

$$Z_1(x) = y_1 + b_1(x - x_1) + c_1(x - x_1)^2 + d_1(x - x_1)^3 \, on[x_1, x_2] \qquad \text{(Eq 7)}$$

$$Z_2(x) = y_2 + b_2(x - x_2) + c_2(x - x_2)^2 + d_2(x - x_2)^3 \, on[x_2, x_3] \qquad \text{(Eq 8)}$$

$$Z_{n-1}(x) = y_{n-1} + b_{n-1}(x - x_{n-1}) + c_{n-1}(x - x_{n-1})^2 + d_{n-1}(x - x_{n-1})^3 \, on[x_{n-1}, x_n] \qquad \text{(Eq 9)}$$

With the following conditions (known as properties)

a. $Z_i(x_i) = y_i$ and $S_i(x_{i+1}) = y_{i+1}$ for $i = 1, \ldots, n - 1$ $\qquad$ (Eq 10)

This property guarantees that the spline $S(x)$ interpolates the data points.

b. $Z'_{i-1}(x_i) = Z'_i(x_i)$ for $i = 2, \ldots, n - 1$ $\qquad$ (Eq 11)

$Z'(x)$ is continuous on the interval $[x_1, x_n]$; this property forces the slopes of the neighboring parts to agree when they meet.

c. $Z''_{i-1}(x_i) = Z''_i(x_i)$ for $i = 2, \ldots, n - 1$ $\qquad$ (Eq 12)

$Z''(x)$ is continuous on the interval $[x_1, x_n]$, which also forces the neighboring spline to have the same curvature, to guarantee smoothness.

## 2.4 Artificial neural networks ANN

Imitate cognitive and theoretical knowledge of how the human brain learns, including neural interconnections in the network. ANN is a fully linked multi-neuron network having an input layer that communicates with the following layer [74–76]. Artificial Neural Networks (ANN) try to mimic the human brain. Humans may be slower and less precise than neural networks [77]. An artificial neural network (ANN) is a fully linked multi-neural network that consists of an input layer, a layer that communicates with the layer below it (hidden layers), a layer that processes additional weighted connections and links to the output layer, and finally, an output layer of nodes that provide a variable for output [78,79]. Neural network nodes output real numbers from inputs. Nodes' outputs are calculated by applying a function ($\psi$) to the previous layer nodes' outputs [80]. Before that, the input layer output ($o(0)$) equals the input. We calculate the output layer output by computing layer outputs sequentially. This is an inferential procedure. Neural networks are structured according to the number of layers and the functions that govern the outputs of each layer.

$$o^{(k)} = \begin{cases} \psi_{(k)}(o^{(k-1)}), & \text{if } k \geq 1 \\ x_n, & k = 0 \end{cases} \qquad \text{(Eq 13)}$$

Where $o^{\{k\}}$ = the output vector representing the outputs of nodes in the layer $l^k$, $o_i^{(k)}$ = the general output of the node $l^k$, $\psi_{(k)}^{(.)}$ = function to determine the outputs of nodes $o_i^{(k)}$ and $x_n =:$ nth input data. In order for two consecutive layers to be fully connected, all nodes in the previous layer must be connected to the next. Fully linked neural network topology is shown in Fig 1.

Neural network predictions are closer to actual standards since the model may be changed iteratively to reduce inaccuracies. Handwriting recognition, medicine, forecasting, and many other fields use ANNs because they outperform conventional technologies [81,82]. Convolutional neural networks (CNNs) are derived from visual cortical neurobiology. It interprets visuals as RGB layers of color and perceives them as three-dimensional entities. It then applies filters on convolutions of inputs in order to extract useful features [83]. In recent times, CNN's achievements in computer vision tasks including as object identification, real-time face recognition, and image categorization and recognition have garnered significant attention [20]. Recurrent Neural Networks (RNN) are feed-forward networks with feedback edges that provide context [84]. ANNs' hidden layers have duplicate cells whose states are modified by past and current feedback input. The ANN's t-1 decisions affect its t-time judgment [50,85]. Cell-internal connection differences determine network design capability. Backpropagate over time with RNN memory, which stores all concealed states. Our networks learned short-range dependencies but not intermediate-range ones. Other scientists have addressed this restriction by allowing RNN to acquire mid-term dependencies Recurrent neural networks may feature a bidirectional, gated, or long-short-term memory architecture (LSTM). The challenge arises when the RNN must store in memory long-term dependencies. Long short-term memory networks solved this problem (LSTM) [86]. The LSTM uses truncation to efficiently flow a range of constant faults in special unit internal states.

## 2.5 Deep learning approach

Deep learning algorithms have outperformed computer vision in recent years, but their time series models are still shallow. Data scientists in most sectors utilize autoregressive models

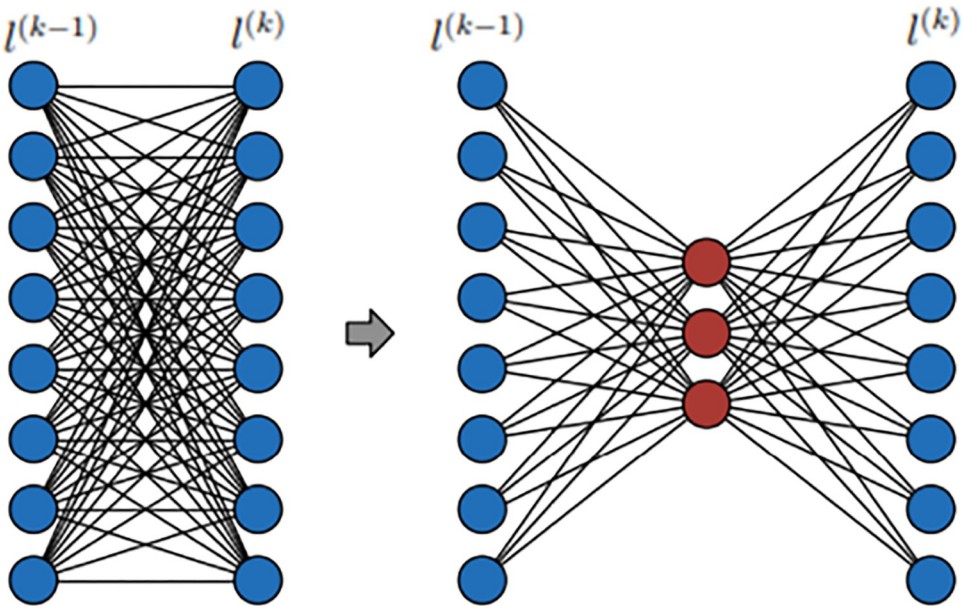

**Fig 1. The graph representation of a fully connected neural network.**

instead of deep learning. Common reasons for using these methods include interpretability, constrained data, convenience of use, and training cost [87]. It typically improves both performance (compared to other vanilla LSTMs/RNNs) and interpretability [88]. Their speed surpasses that of RNN/LSTM in a few of cases [89]. In addition, attention networks include categorical inference in deep neural networks [90]. These brain networks control consciousness, sensory orientation, and competing reactions. Many sequence transduction models use complex recurrent or convolutional neural networks in an encoder-decoder arrangement, but they only work locally and lack global information [91]. This constraint prompted attention networks to record long-range interactions. Attention networks are popular due to their parallel computation and relationship modeling adaptability. A simple residual network model that replaced all spatial convolutions with self-attention showed attention networks' impact. This fully self-attentional model surpassed the baseline with less categorization parameters [92]. Additionally, subsequent tests on attention in general deep neural networks show continuous progress in categorization and detection with varied models. This influenced this study's use of attention networks [93].

## 2.6 RNN & LSTM

Recurrent Neural Networks (RNNs) and Long Short-Term Memory (LSTM) models exhibit remarkable capabilities in processing sequential data and uncover complex relationships that are concealed within time series. Long-sufficiency-term (LSTM) models, classified as a subtype of RNNs, exhibit exceptional efficacy in managing data dependencies that endure for an extended duration. Furthermore, the greatest distinction between RNN and other networks is that it shows the relationship between current and historical information, although gradient dispersion limits its ability to learn relevant content quickly. LSTM (Long short-term memory) was introduced in 1997 to address the vanishing gradient issue [94]. Unlike the standard RNN, the LSTM has the same repeated modules, but its underlying structure is three "Gates" instead of a single neural network layer, making it better for processing and predicting critical events

with lengthy intervals and delays in time series [95]. RNNs are crucial to deep learning algorithms that operate on time-series data. Remembering historical data allows these neural networks to predict current events. [96]. Due to fading and expanding gradients, conventional RNNs struggle to manage long-term dependencies in extended time series. After developing the LSTM network, critical state information was extracted from historical time-series data [97]. This event reinforces the link between past and future data. LSTM has been effective in welding quality monitoring and penetration prediction [98].

Another major perk is the TSA-LSTM model's ability to process massive amounts of time-series data quickly without sacrificing forecast accuracy [99]. Cancer researchers and practitioners can benefit greatly from its sophisticated design and capacity to simulate intricate temporal patterns, which enhances their predicting abilities and allows them to deliver superior patient treatment [100]. However, unlike RNN, LSTM allows just one memory superposition formula. Show two network diagrams. Data analysis and modeling are done using the LSTM neural network for long-term time series data [101]. Long Short-Term Memory RNNs are efficient in deep learning. LSTM temporal model emphasizes keyhole progression during welding. The LSTM network learns temporal feature information from image sequences, demonstrating keyhole dynamics [102]. LSTMs use memory cells instead of artificial neurons in the hidden layer, unlike feedforward neural networks [103–105]. Also, systems can rapidly correlate memories and input remotely with memory cells to understand and predict data structure over time. Internal indirect state units are LSTM long-term or short-term memory. A gate to the conventional recurrent cell improves retention for "long-term dependencies" in the LSTM cell. Researchers used this to produce LSTM topologies with and without forget gates [106,107]. In the LSTM network, indirect state units function as either short-term or long-term memory. Unit status regulates the generation of an LSTM network. This is not only the last entry; it is a fundamental quality when a network forecast must be influenced by input history. It processes video, audio, and image data [108].

In the end, the TSA-LSTM is an advanced deep-learning architecture used in predictive analytics to predict cancer incidence rates. This approach exploits LSTM networks' ability to capture long-term associations in sequential data. The model utilizes LSTM networks, known for their proficiency in processing sequential data such as cancer development, enabling the model to grasp the evolving characteristics of the disease. Furthermore, the two-stage architecture has the potential to learn intricate correlations among various elements in the data. The initial phase extracts fundamental patterns, which are then utilized in the subsequent phase for prediction or classification purposes. At various levels of prediction, attention processes focus on key traits [109,110]. This study used TSLM to predict cancer, which has two stages: Feature extraction first. Use LSTM layers to uncover temporal correlations and sequential patterns from cancer incidence data [111]. After two steps, the model accurately forecasts cancer incidence rates by capturing their complex dynamics. Previous research and articles on TSA-LSTM models for cancer incidence prediction may inform technical implementation [112,113]. The following section describes this model's cancer prediction benefits. We provide historical cancer incidence data, which contains demographics, environmental variables, cancer diagnosis rate time series, and other essential information, unlike other studies. Normalize, fix missing values, and create LSTM-friendly data windows [114]. The TSA-LSTM model captures sequential data long-term dependencies with several LSTM cell layers. LSTM prediction uses attention to highlight relevant traits. Identify and refine important qualities for successful forecasting in two steps. Compare predicted values against real-world findings to assess the model's accuracy [115]. Assess model estimates using epidemiology, historical patterns, and cancer risk indicators. Attention processes and LSTM networks in a dual-stage framework assess cancer incidence and capture its complex temporal patterns in TSA-LSTM [116].

Through enhanced deep learning, researchers and data scientists can improve cancer risk assessment, resource distribution, and public health policy [117]. This section has comprehensively examined the emergence of big data analytics within enterprises, as well as its application in the healthcare industry [118]. This article examines cancer analytics data visualization. The varied statistical models, deep learning models, and architectures used to anticipate data throughout time are also examined, along with the obstacles that spurred our research.

## 3. Empirical methodology

### 3.1 Data source and experimental manipulation

The dataset came from the Centers for Disease Control and prevention (CDC) and the American Cancer Society (S1 Appendix). The data includes cancer incidence rates, risk factors such as tobacco smoking, alcohol consumption, physical inactivity, obesity/excess body weight, and cancers caused by these risk factors. Our 1990–2021 data spans 32 years. Moreover, the study sets itself apart by developing a methodology to forecast the occurrence of cancer by the year 2025. The deployment of a novel two-stage attention model is used to forecast the rate of growth in cancer incidence, with 2017 serving as the reference year. It has been established that the prevalence of cancer is projected to increase by 9 percent by December 2020 and 12 percent by December 2021. There is an expected increase of 14 percent and 16 percent in newly diagnosed cancer cases by the end of 2022 and 2023, respectively. Furthermore, it is estimated that the number of cancer cases will rise by 19 percent and 21 percent in the years 2024 and 2025, respectively. The expected increase in cancer rates can be ascribed, partially, to a rise in detrimental lifestyle choices, such as the consumption of alcohol and tobacco, lack of physical activity, and a subpar diet. These factors contribute to the high occurrence of obesity in the United States. The accuracy and generalizability of a methodology are significantly influenced by the quality and completeness of the training data, in the case of data dependence. Inconsistencies, anomalies, and missing data points can all hinder a model's ability to discover accurate patterns. During the feature selection phase for the TSA-LSTM model, the most pertinent variables or characteristics that are probable to exert a substantial influence on the prediction of cancer incidence and mortality rates are identified. By eliminating extraneous complexity and noise and concentrating on the most informative data points, this procedure is vital for optimizing the model's performance. Furthermore, the complexity of the model. While simpler models may offer greater interpretability, they may fail to encompass the complete intricacy of the data. Extremely complex models, on the other hand, are susceptible to overfitting and have difficulty generalizing to unobserved data. Moreover, with regard to generalizability, it is frequently constructed and evaluated on particular datasets. The TSA-LSTM model selects features by extracting relevant variables from big datasets and using domain expertise to find those most likely to be associated with cancer outcomes. Environmental factors (pollution, carcinogen exposure), genetic markers, lifestyle factors (smoking, nutrition), demographic data (age, gender, ethnicity), and prior cancer incidence rates are commonly considered for feature selection [31,119]. It also protects Americans from international and domestic health, safety, and security risks with special reaction teams to fight diseases [120,121]. It oversees public health trend research.

The American Cancer Society's mission is to save and celebrate lives and fight for a world without cancer. The organization estimates the present cancer burden by projecting the number of new cancer cases and deaths expected each year as cancer incidence and mortality data are three to four years behind [122,123]. Researching and pursuing public health trends is its responsibility. The American Cancer Society is a nonprofit organization dedicated to eliminating cancer and promoting a future without it. Additionally, they save and celebrate lives. Since

cancer incidence and mortality data are three to four years behind the current year, the organization estimates the current cancer burden by projecting future cancer cases and deaths. The TSA-LSTM model consistently shows better predictive performance metrics, such as accuracy, precision, recall, and F1 score, compared to traditional models like logistic regression, decision trees, random forests, and simpler LSTM architectures. The TSA-LSTM model excels in cancer prediction tasks by effectively capturing long-range dependencies and temporal linkages, leading to superior accuracy and reliability compared to other models. Accurate cancer incidence and mortality rate estimates are crucial for healthcare planning and patient outcomes due to the significant impact of early diagnosis and exact forecasting [124,125]. In this regard, we wish to illustrate the correlation between cancer incidence and risk factors including obesity, smoking, drinking, and inactivity. In order to achieve our visualization objective, four preprocessing procedures are necessary. First, we verify each lifestyle choice and its annual cancer incidence in Excel for missing data. In R, we use mean imputation to fill missing values. Missing values are imputed with the sample mean. Third, we delete duplicate observations using Excel. Fourth, we classify lifestyle choices by age, sex, or type of cancer. Finally, for correlation analysis, we use R to sum the average classification data.

The study aims to create a more accurate forecasting algorithm. Our initial cancer data comprises just 32 yearly entries (February 1990 to March 2021), which is not enough to improve model training and prediction. To improve outcomes, the study interpolates data into numerous points. Use interpolation to discover the optimum data enhancement method. Interpolation is estimating an unknown quantity between two known variables or inferring missing information from one. Building a function that intersects a discrete set of known data points is the core of this strategy. When missing data is available and cycles are long, this technique works [126,127]. A variable's predictive strength, relevance to the result, and multicollinearity concerns usually determine its inclusion. When variables are highly correlated with the target variable and can illuminate patterns, they are more likely to be included in the model. Superfluous variables that don't affect predictions or add noise may be deleted to simplify the model. Through domain knowledge-based pruning, correlation analysis, feature importance ranking, and mutual information analysis, TSA-LSTM model feature selection variables can be identified. The feature selection method is critical for ensuring that the TSA-LSTM model can accurately capture the complex correlations in cancer data and to optimize its performance. Constantly, it is assumed that the performance of the model will effectively extrapolate to unobserved data that possesses comparable attributes. Nevertheless, such an outcome cannot always be ensured, particularly in cases where the unobserved data diverges substantially from the training data. The model's ability to generate more reliable estimates of cancer incidence and mortality rates is due to its rigorous selection process that eliminates extraneous or redundant components.

A frequent interpolation method is spline. Spline interpolation uses a piecewise polynomial as the interpolant. Many prefer spline interpolation over polynomial interpolation because the inaccuracy can be minimized even with low-degree polynomials. Spline interpolation eliminates Runge's phenomenon, which causes oscillation between locations when using high-degree polynomials [28,123,128]. As in 2.4, spline interpolation can be linear, quadratic, or cubic. We purposely construct missing values in 32 data points to decide the study's interpolation strategy (Linear, Quadratic, or Cubic). We keep only certain years' data (1990, 1994, 1997, 2001, 2004, 2007, 2010, 2012,2014,2017,2019 and 2021). We now utilize each interpolation method to predict and fill in missing data for other years (1991, 1992, 1993, 1995, 1996, 1998, 1999, 2000, 2002, 2003, 2005, 2006, 2008, 2009, 2011 2013,2015,2017,2019, and 2021) Interpolation, which anticipates missing figures more accurately, is our choice. Each method's missing value-filling results are in (S2 Appendix) The table shows that all three models predicted but

did not match. Therefore, we augmented our data using the cubic spline technique which upgraded and increased the data into hundreds of entries [41,113]. This folded the data from yearly to monthly entries, using some for-model testing and the rest for training. Adding synthetic data points to our data would improve model training for accurate deployment. The data is augmented using the cubic spline. Focusing on our raw data (from 1990–2021), we have 32 data points for the cubic spine for 420 data points.

## 3.2 Variable construction and correlation design

Tableau will be used to highlight the association between lifestyle and cancer incidence. Our dependent variable new cases, fatalities, and cancers are linked to lifestyle choices. Smoking, alcohol, nutrition, inactivity/lack of exercise, and bad nutrition cause obesity. In this study, cancer data shows a dependent variable, where the total number of new cancer incidences per year. Tobacco smoking, alcohol consumption, physical inactivity, and obesity show the independent variables.

## 3.3 Tableau data visualization

Effectively using people and data is as important as the method. Data must be reviewed, cleaned, processed, and modeled to find usable information, draw conclusions, and aid decision-making [31,129].Tableau is a powerful intelligence platform that uses analytics to harness data. It streamlines data viewing, processing, and comprehension. New data modeling tools in Tableau make sophisticated data analysis easier without database knowledge. We can seamlessly mix data utilizing table relationships to maintain the data model adaptable for analysis [130,131]. We describe forecasting, analysis modeling, statistical measures, and smoothing coefficients dataset processing. For comprehension, Tableau visualizes data. Launch. Tableau's launch page links to databases, text files, and other data. Tableau analysis and source access begin. Once connected and imported, Tableau instantly displays all associated data sets on the data source display page. Without unnecessary data, the data translator cleans and visualizes loaded data on this page. Create Tableau visualizations and spreadsheets using all loaded data. Create dashboard and story visualizations by dragging and dropping measures and dimensions in Tableau's spreadsheet. Enhancing perspectives Dashboards should make sense from data visualization. Dashboards with many stories allow users to make crucial decisions. In Tableau dashboards, data can be dragged and dropped into stories. [31,125,127]. In the results and analysis, many visualizations are discussed. LSTMs are neural network architectures, not models, therefore they can be changed for specific operations. A rudimentary LSTM model may perform poorly due to weak operations. The basic LSTM model presented to handle gradient issues (Exploring Gradient Problems and Vanishing Gradient Problems) contained simply an input and output gate and maintained the RNN's cell state [113]. Besides gradient difficulties, the RNN had input and output weight conflicts that the simple LSTM solved. Though better than the RNN, this technique does not fully solve its problems [113,132]. A forget gate reduced the number of sequences truncated in the original LSTM Model by deleting valuable material from the hidden state and forgetting unneeded ones. Forget-gate biases impact LSTM performance. Many academics have presented Vanilla, Stacked, CNN, Bidirectional, and other LSTM designs. Evolutionary approaches were preferred over gradient descent by Shalev-Shwartz, Shamir, and Shammah (2017). Attention emphasizes certain data [23,133]. Studies have shown that attention networks are needed to exhibit data parts. This study proposes a two-stage attention model to improve the LSTM model's attentiveness and prediction accuracy [28,123]. Our work visualizes lifestyle patterns and specific malignancies and estimates future cancer incidences using raw data cleanup and sample enlargement. Interpretive

research and an innovative way to examine complicated lifestyle datasets with Tableau dashboard visual analytics were also discussed. The chapter explained our data preparation materials and new model design. This part also checks our data interpolation strategy for optimal results.

# 4. Results and analysis of modelling based on LSTM

## 4.1 Double exponential smoothing data analysis modeling and forecasting

Here, we examine and describe the data analytic modeling used to anticipate cancer data using Tableau's double exponential smoothing approach. Alpha and beta values of 0.5 produced the lowest error margin in LSTM-based data processing and analysis models. In testing, these coefficients were employed. Table 1 shows forecast alternatives based on actual yearly time series entries from 1990 to 2021 and measured by age-standardized measure per 100,000.

The forecast is extended to three years without seasonal pattern, although the model seeks a recurrent seasonal pattern in every period [134]. The forecast model shows multiplicative levels, trends, and no seasonality. Root mean squared error (RMSE), Akaike information criterion (AIC), mean absolute scaled error (MASE), mean absolute percentage error (MAPE), and mean absolute error (MAE) are used to ensure forecast quality in DES [135,136]. Each quality indicator provides statistical model quality data. Level, trend, and seasonal smoothing coefficients are alpha, beta, and gamma. Level and trend smoothing coefficients alpha and beta are 0.50 and gamma is 0.00 for the forecast. Forecasts are more naïve because to smoothing coefficients in our investigation. In this study when assessing the effectiveness of a prediction technique, it is essential to consider more than just one measurement. Two important approaches, Root Mean Squared Error (RMSE) and R-squared ($R^2$), provide essential information's primarily measures the size of mistakes, giving the average proximity between predictions and actual values (lower RMSE indicates better accuracy). The coefficient of determination, $R^2$, quantifies the degree to which the model's predictions align with the general pattern observed in the data (a greater $R^2$ indicates a stronger fit). By utilizing both measurements, one can acquire a fuller comprehension of the model's performance—its average accuracy and its ability to effectively capture the overall patterns. Furthermore, the use of boxplots to visualize the distribution of errors is quite useful. Boxplots can expose outliers or biases in the predictions,

**Table 1. Summary of forecast options and model used in DES.**

| Time Series | Year | | | |
|---|---|---|---|---|
| Measures | Sum of Age-standardized (per 100.00) | | | |
| Forecast Length: | 3-year periods (February 2019- March 2021) | | | |
| Forecast based on: | February 1990 –March 2021 | | | |
| Ignore last: | No periods ignored | | | |
| Seasonal Pattern Prediction Interval | None 100% | | | |

**Summary**

| Initial 2014 | Δ from Initial (2015–2021) | Seasonal Effect | | Contribution | | Quality (0 if False, 1 if True) |
|---|---|---|---|---|---|---|
| | | *High* | *Low* | *Trend* | *Season* | |
| 136.95 ∓ 4.40 | 3.86 | None | | 100% | 0.0% | 1 (Good) |

**Models**

| *Model* | | | *Quality Metrics* | | | | | *Smoothing Coefficients* | | |
|---|---|---|---|---|---|---|---|---|---|---|
| **Level** | **Trend** | **Season** | **RMSE** | **MAE** | **MASE** | **MAPE** | **AIC** | **α** | **β** | **γ** |
| Multiplicative | Multiplicative | None | 6.513 | 3.525 | 0.08 | 0.0% | 6.291 | 0.71 | 0.62 | 0.000 |

offering a more comprehensive examination beyond the average error measured by RMSE. Overall, the performance of a prediction approach can be thoroughly assessed by using a combination of RMSE, $R^2$ (Coefficient of Determination), and boxplots.

## 4.2 TSA-LSTM model design using LSTM

The paper's model is based on time series lifestyle and cancer incidence data from the Centers for Disease Control and Prevention and the American Cancer Association from 1990–2021. Thus, it is better for LSTM network applications since it solves the complexity and timeliness issues produced by the three gates. First and foremost, the TSA-LSTM model attention mechanism prioritizes the input characteristics, specifically their significance, to guarantee that the model encoder converges to specific characteristics of the input sequence when predicting the specific characteristics of the output sequence. This enhances the model's natural learning trend and prediction quality. The quality of predictions is enhanced by selecting network features and maintaining model performance in real-time. Our model employs a Long Short-Term Memory (LSTM) Recurrent Neural Network that is capable of training long-term dependencies. LSTMs avoid long-term reliance, making long-term recall possible [23,113]. We exploited attention mechanisms and their strong operations to retain major cancer trend features for a decent forecasting model for active cancer research. The network uses attention processes to focus on one part of a complex input until classification. To simplify difficult tasks, process smaller regions sequentially. Like the human mind breaks down a new difficulty into smaller tasks and addresses them separately [41,127]. Input attention, temporal attention, and repetition help our model outperform baseline models in studies. Based on the model idea, we propose the Two-Stage Attention LSTM (TSA-LSTM) model for cancer data prediction in this research [137]. Sensitivity analysis helps users understand the model's sensitivity to a wide range of scenarios and potential sources of uncertainty. This improves user decision-making. These goals are achieved by perturbing input features and analyzing the results. Healthcare practitioners can develop trust in the model and understand its restrictions by providing model-specific documentation on its structure, training methods, input characteristics, and output assessments. These techniques increase the TSA-LSTM model's interpretability, making it accessible to healthcare practitioners without machine learning experience [23,35].

The following sections of this section provide a detailed explanation of the model's components and provide essential insights into its testing configuration. In this work, we suggest a time series model for the accurate evaluation and long-term prediction of cancer incidences. Given $n$ and $\tau$ as input sequence and window size length respectively; the yearly incidence movement is denoted $\mathbf{X} = (\mathbf{x}^1, \mathbf{x}^2, \mathbf{x}^3, \ldots, \mathbf{x}^n)^{\mathrm{T}} = (\mathbf{x}_1, \mathbf{x}_2, \mathbf{x}_3, \ldots, \mathbf{x}_n) \in \mathbb{R}^{n \times \tau}$. Then, the moving sequence of length $\tau$ is denoted $\mathbf{x}^a = (x_1^a, x_2^a, x_3^a, \ldots, x_\tau^a)^{\mathrm{T}} \in \mathbb{R}^{\tau}$ an, d a vector of $n$ moving input sequence at time $t$ as $\mathbf{x} = (x_t^1, x_t^2, x_t^3, \ldots, x_t^n)^{\mathrm{T}} \in \mathbb{R}^n$. Again, given the previous values of the target sequence as $(y_1, y_2, y_3, \ldots, y_{\tau-1})$ and $y_t \in \mathbb{R}$, and the current and past values of the $n$ moving sequence as $(\mathbf{x}_1, \mathbf{x}_2, \mathbf{x}_3, \ldots, \mathbf{x}_\tau); \mathbf{x}_t \in \mathbb{R}^n$, non-linear autoregressive exogenous models aim to optimize nonlinear mappings to the current value of a target sequence $y_\tau$ using $\hat{y}_\tau = F(y_1, y_2, \ldots, y_\tau, \mathbf{x}_1, \mathbf{x}_2, \ldots, \mathbf{x}_\tau)$ with $F(\bullet)$ as the non-linear mapping function to determine via optimization. In the suggested model, we used LSTM units as encoding units [127,138]. We chose an LSTM encoding unit because its cell states sustain states across time, avoiding vanishing gradients and improving long-term information retention in our time series model. Formally, taking each time $t$, an input vector $\mathbf{x}_t$, a memory state vector $\mathbf{c}_t$, and $\mathbf{h}_t$ as the output of $\mathbf{c}_t$, then the LSTM units can be formulated as: where, $\mathbf{W}$ and $\mathbf{U}$ are the weight matrices. $\mathbf{b}$ is a bias vector. $\mathbf{i}_t, \mathbf{f}_t, \mathbf{o}_t$ are the input, forget, and output gating units and $\sigma$ is our activation function for the gating units. For our time series prediction at the time $t$, given the input sequence

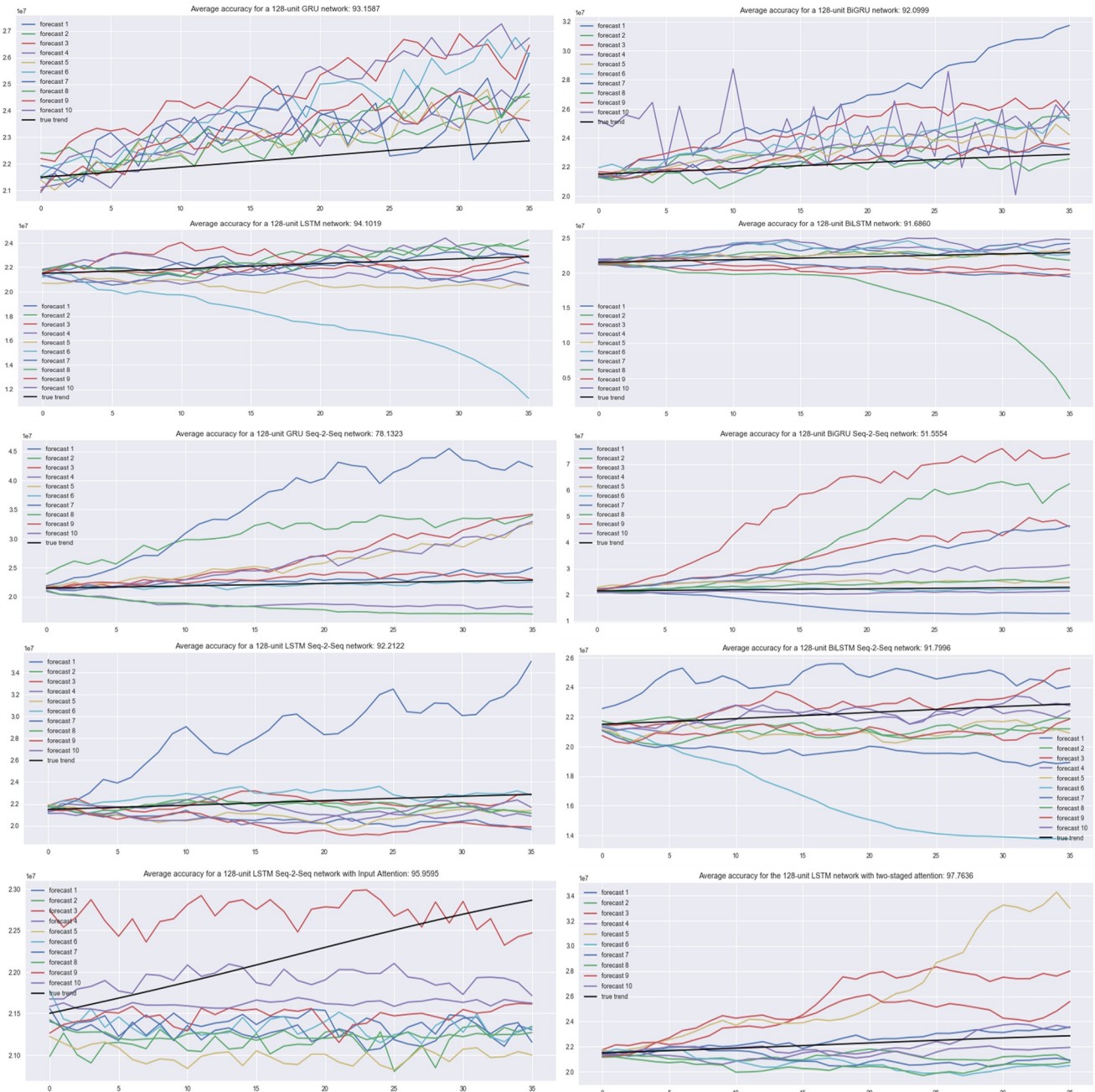

**Fig 2. TSA-LSTM design.**

$\mathbf{X} = (\mathbf{x}_1, x_2, x_3, \ldots, x_\tau) | \mathbf{x}_t \in \mathbb{R}^n$, we apply our LSTM encoder to learn the mapping optimum mapping from $\mathbf{x}_t$ to $\mathbf{h}_t$ using $\mathbf{h}_t = \mathbf{f}_1(\mathbf{h}_{t-1}, \mathbf{x}_t)$ $\mathbf{h}_t \in \mathbb{R}^s$, where $\mathbf{h}_t$ is the hidden state of the encoder at the time $t$ $\mathbf{f}_1$ is our LSTM non-linear activation function $s$ is the size of the hidden state $\mathbf{h}_t$.

The existing models are compared in (Fig 2) based on their average accuracy ratings in testing and forecasting the cancer data. We trained and evaluated each model ten times in the same environment. Each model overwrote the training losses of the previously implemented models. The initial testing of the Gated Recurrent Unit model yielded an average accuracy of 94.27% for 128 units. The first tested network, the GRU network, showed a clear intent with

an accuracy of 93.158. The second indicator is the BIGRU network (92.099), which displays 94.14%. Three, the LSTM network (94.101) exhibits a 79.23% percentage. Following this, the BILSTM network (91.686) exhibits a 54.87% increase. Fifth, the GRU seq-2 network (78.132) displays a 93.78% accuracy. The sixth network, the BIGRU seq-2 network (51.555), displays a 93.43%. Seventh, the LSTM seq-2 network (92.212) exhibits a 93.38% rating. Eighth, the BILSTM seq-2 network (91.799) boasts a 95.98% efficiency. On the ninth position, the LSTM seq-2-seq network with input intention (95.959) exhibits an accuracy of 87.98%. The LSTM network with two stages of attention (97.763) demonstrates a 98.87% accuracy the tenth time. Finally, the testing set executes the proposed TSA-LSTM model. Each model overwrote the training losses of the previously implemented models. Results indicate an average accuracy of 87.96%.

## 4.3 Two stage attention mechanism

Our LSTM encoder reads and encodes input sequences into fixed-length vectors for two-stage attention. Sequences encoded as fixed-length vectors prohibit models from memorizing longer ones (Eqs 14–18). We augment our model's encoding unit to boost prediction accuracy. The model's LSTM encoder's convergence on input and output sequence features is examined first. Learning improved model prediction. We construct a second attention mechanism, temporal attention, to choose critical network attributes at all times while decoding to preserve model performance as sequence length rises [125,139]. Real-time model performance picks network features and improves prediction. Assuming a-th moving sequence input, $\mathbf{x}^a = (x_1^a, x_2^a, x_3^a, \ldots, x_\tau^a)^{\mathrm{T}} \in \mathbb{R}^\tau$, we create an input attention mechanism with a multilayer perceptron by applying the LSTM cell state $\mathbf{c}_{t-1}$ and the previous hidden state $\mathbf{h}_{t-1}$ in the equation:

$$e_t^a = \mathbf{v}_e^{\mathrm{T}} \tanh(\mathbf{W}_e[\mathbf{h}_{t-1}, \mathbf{c}_{t-1}] + \mathbf{U}_e \mathbf{x}^k) | \alpha_t^a = \exp(e_t^a) \left[\sum_{i=1}^n \exp(e_t^a)\right]^{-1} \qquad \text{(Eq 14)}$$

Where

- $\alpha_t^a$ is the attention weight that holds the relevance of the $a$-th input feature at time t.

- $\mathbf{v}_e^{\mathrm{T}} \in \mathbb{R}^\tau$, $\mathbf{W}_e \in \mathbb{R}^{\tau \times 2s}$, $\mathbf{U}_e \in \mathbb{R}^{\tau \times \tau}$ are learnable parameters.

Lastly, we sum all the attention weights to 1 by applying a soft auction to $e_t^a$. In combining the attention mechanism with the LSTM encoder, a resulting sequence, $\hat{\mathbf{x}}_t = (\alpha_t^1 x_t^1, \alpha_t^2 x_t^2, \alpha_t^3 x_t^3, \ldots, \alpha_t^n x_t^n)^{\mathrm{T}}$, the hidden state of the encoder is updated as $\hat{\mathbf{h}}_t = \mathbf{f}_1(\mathbf{h}_{t-1}, \hat{\mathbf{x}}_t)$ and thus, allows the LSTM encoding unit of the model can selectively focus on unique characteristics of the input sequence. We construct an LSTM decoding unit that uses the model's internal representation to predict output $\hat{y}_\tau$. We construct an LSTM decoding unit that uses the model's internal representation to predict output a. In order to maintain the performance of the model through the entire network with as the sequence length increases, we construct a temporal attention mechanism that chooses vital features of the network at all times while decoding. Given that at time $t$, the attention weight of an encoder's hidden state is computed as:

$$l_t^i = \mathbf{v}_d^{\mathrm{T}} \tanh(\mathbf{W}_d[\mathbf{d}_{t-1}; \hat{\mathbf{c}}_{t-1}] + \mathbf{U}_d \mathbf{h}_i) | 1 \leq i \leq \tau \qquad \text{(Eq 15)}$$

with

$$\beta_t^i = \exp(l_t^i) \left[\sum_{i=1}^\tau \exp(l_t^i)\right]^{-1} \qquad \text{(Eq 16)}$$

where

- $\mathbf{d}_{t-1} \in \mathbb{R}^p$ is the hidden state of a previous decoder.

- $\hat{\mathbf{c}}_{t-1} \in \mathbb{R}^p$ is the cell state of the LSTM unit.

- $\mathbf{v}_d^{\mathrm{T}} \in \mathbb{R}^s$, $\mathbf{W}_d \in \mathbb{R}^{s \times 2p}$, $\mathbf{U}_d \in \mathbb{R}^{s \times s}$ are learnable parameters.

Similar to the encoding unit's input attention mechanism, the temporal attention mechanism holds the relevant features, but this time, of the $i$-th encoder hidden state for prediction using attention weight $\beta_t^i$ at the decoding level [140]. Considering the encoder's hidden state $\mathbf{h}_i$ is mapped to the input's temporal component, the attention mechanism evaluates the context vector $\mathbf{c}_t$ (unique at each time) as the weighted sum of all the hidden states $\{\mathbf{h}_1, \mathbf{h}_2, \mathbf{h}_3, \ldots, \mathbf{h}_\tau\}$ of the encoding units:

$$\hat{\mathbf{c}}_t = \sum_{i=1}^{\tau} \beta_t^i \mathbf{h}_i \qquad (\text{Eq 17})$$

Hence, the combination of the target sequence and context vectors arrives at:

$$\hat{y}_{t-1} = \hat{\mathbf{w}}^{\mathrm{T}} [y_{t-1}; \mathbf{c}_{t-1}] + \hat{b} \qquad (\text{Eq 18})$$

where $\hat{\mathbf{w}} \in \mathbb{R}^{m+1}$ and $\hat{b} \in \mathbb{R}$ maps $[y_{t-1}; \mathbf{c}_{t-1}]$ to the decoder's size. $\hat{y}_{t-1}$ is then used to update the decoder's hidden state at time $t$. We created and implemented nine neural baseline models to compare our model to others using neural networks. Our hierarchical neural model has customizable window and encoder/decoder hidden state sizes. We chose the two-stage attention encoder-decoder model since it obtained the best results among state-of-the-art models. The TSA-LSTM model pseudocode is in S3 Appendix. The analysis and correlational data have been refined with the inclusion of six subfactors. Priority is given to the following associations: alcohol use and cancer, tobacco use and cancer, obesity and cancer, physical inactivity and cancer, and cancer itself. The sixth proportion of malignancies in the United States that are ascribed to risk factors exhibits temporal patterns of incidence. Female breast cancer is one of the most common alcohol-related cancers in the US, with 138.3 per 100,000 cases [41,141]. Women who drink excessively are at risk of breast cancer. Alcohol intake is also linked to 65.8 colon and rectum cancers, 21.7 lip cancers, 8.1 liver cancers, 5.8 esophageal cancers, and 3.3 larynx cancers per 100,000 persons This visual uses a horizontal polygon map to quickly visualize data patterns. Second, (Fig 3) depicts how tobacco smoking promotes cancer. Tracheal, lung, bronchus, colon rectal, and urinary bladder cancers are the most common tobacco-related malignancies. Smoking causes 58.80 cases of tracheal, lung, and bronchus cancer per 100,000 persons, making it the most common cancer caused by smoking. Tobacco smoking causes 49.70 colon and rectal cancers, 30.10 urinary bladder cancers, and 36.60 kidney/renal pelvis cancers. Additionally, tobacco smoking causes 14.20 cases of acute myeloid leukemia per 100,000 people. Additionally, the cancer is the most terrifying outcome of smoking, which casts a dark cloud over people's health. And smoke increases the chance for a terrifying array of malignancies due to its over 7,000 compounds, including 70 proven carcinogens, which rampage through the body [142]. There is extensive harm, including but not limited to lung cancer (the leading cause of cancer-related deaths, accounting for 80% of all cases) and malignancies of the esophagus, larynx, lip, mouth, pharynx, bladder, and kidneys [143].

Third, the bubble chart in (Fig 3) illustrates that obesity is a major cause of cancer. Breast cancer after menopause is the most common cancer-related obesity, followed by colon, rectum, uterine, thyroid, pancreatic, ovary, and kidney malignancies. Obesity is linked to 439 postmenopausal female breast cancer, 43.9 colon/rectum cancer, 46.7 corpus/uterine cancer, and 31.4 thyroid cancers per 100,000 cases. Due to obesity, 21.2 new pancreatic, 31.2 ovarian,

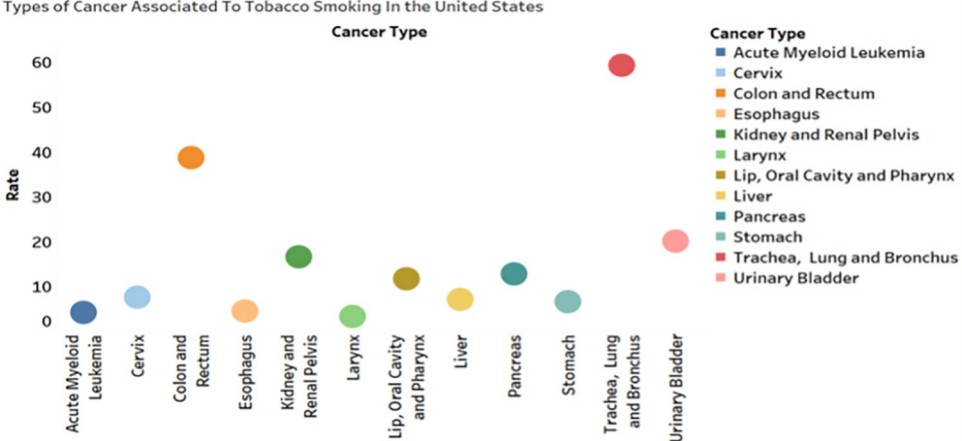

**Fig 3. Cancers attributable to tobacco smoking.**

and 40.9 kidney malignancies are diagnosed. Obesity is linked to liver, gallbladder, esophageal, and meninges cancers [123,144]. The smoke can reach every part of the body, including the digestive tract. It can alter the balance of good bacteria in the gut, which can lead to colon cancer, as well as other malignancies in the liver and stomach. Smoking raises the risk of Acute Myeloid Leukemia, while the link is less clear [145]. Cancer risk drops considerably after the body's extraordinary healing mechanisms take effect. When people stop smoking, they reclaim agency, enhance air quality, and dramatically reduce their cancer risk. Color and size of circles demonstrate how fat impacts cancer [146]. Attribution increases with circle size. The bubble chart is visually appealing and makes it easier to identify categories and compare obesity attribution data by bubble size. The aspect of interpretability holds significant importance, particularly within healthcare environments where the outcomes of patient outcomes can be profoundly impacted by decisions predicated on predictive models [18,127]. The TSA-LSTM model is difficult, however healthcare professionals without machine learning model knowledge can apply many approaches to simplify its conclusions. Essentiality of the feature Visualisations or quick summaries of the model's essential characteristics may help healthcare providers[20]. During the process of prediction, the visualization of the attention mechanism that is utilized by the TSA-LSTM model might provide information regarding which particular segments of the input data are given priority by the model. In the process of making judgments, this can be of assistance to users in understanding the manner in which the model assigns weights to different time steps or qualities [22]. Explaining forecasts may help healthcare providers understand the model's logic for predicting a patient or case.

Fourth, physical inactivity and cancer show (Fig 4) that physical inactivity is the leading cause of US female breast, colon, and corpus/uterine cancers. Physical inactivity causes 438.2 postmenopausal female breast cancers per 100,000 persons. Physical inactivity also causes 38.6 colon, rectum, and 35.7 corpus/uterine cancers in men and women. If female youngsters exercise more, the worrying number of female breast cancers may decrease [147]. It is possible that the present graph, which attempts to demonstrate a correlation between obesity and cancer incidence, is deceiving. It probably shows the total number of cancer cases, but it doesn't show how much of an increase in risk there is because of fat [148]. This is due to the fact that, similar to breast cancer, a high total incidence does not always indicate that a large percentage of those instances are attributed to obesity. The ideal representation would center on the percentage of cancer cases linked to obesity for each kind in order to offer a more accurate picture.

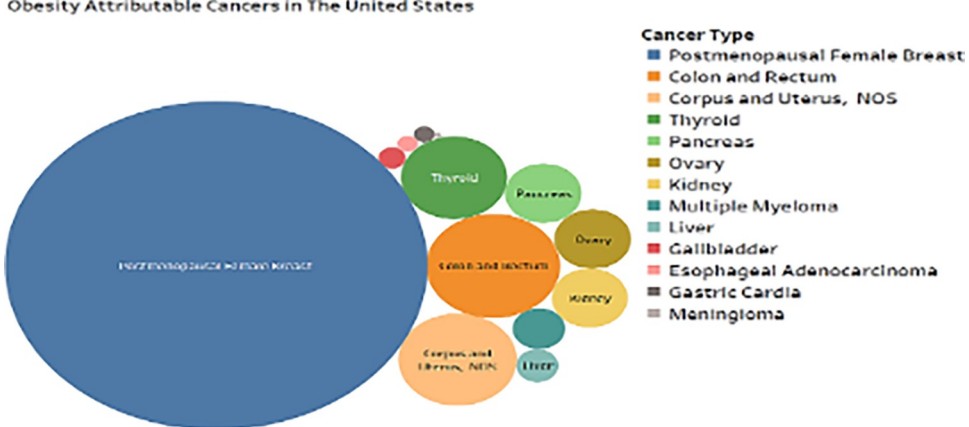

**Fig 4. Cancer-related obesity in the USA.**

Alternatives include stacked bars, proportionate bars, and heatmaps. A heatmap would use color intensity to show attributable risk, proportionate bars to show obesity's percentage of cases, and stacked bars to divide the total incidence into obesity, other risk factors, and unknown causes. Consider age and sex for a clearer image[149]. Menopause may affect breast cancer obesity risk factors. The graphic should be accompanied by the number of obesity-related cancer cases for clarity. Focusing on these regions helps researchers understand obesity's particular impact in cancer risk.

Fig 5 bar chart simplifies comparing malignancies caused by physical inactivity and speeds up decision-making. Fifth, US cancer incidence statistics from 1990 to 2021 reveal that cancer incidence for all ages is rising, indicating a bigger number of cancers. New cancer cases soared to approximately 23 million. Cancer rates worsened and increased with time [150]. The line chart in this study simplifies these perspectives and trends. Sixth, Fig 6's bar chart reveals that

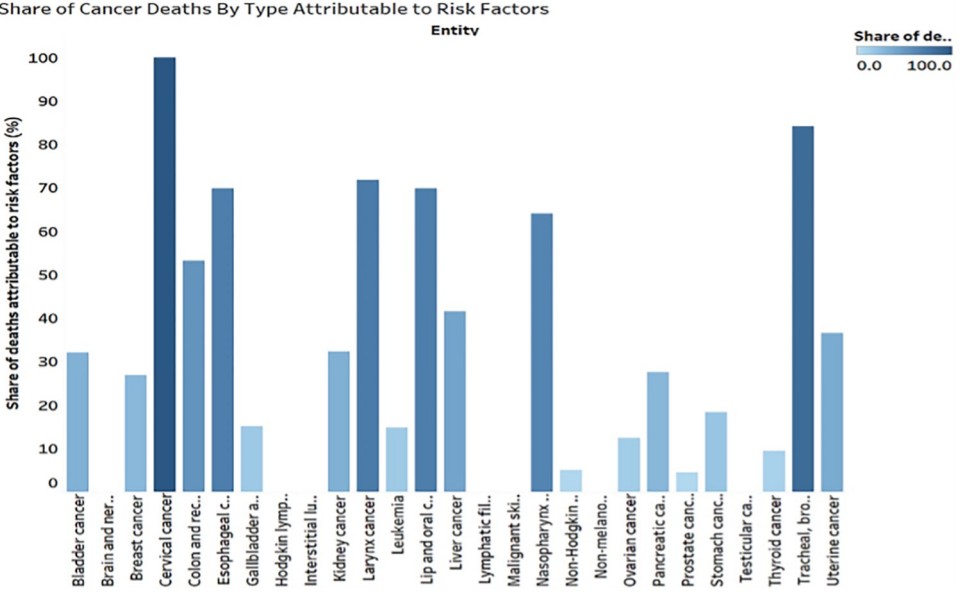

**Fig 5. Share of cancer deaths by type attributable to risk factors in the USA.**

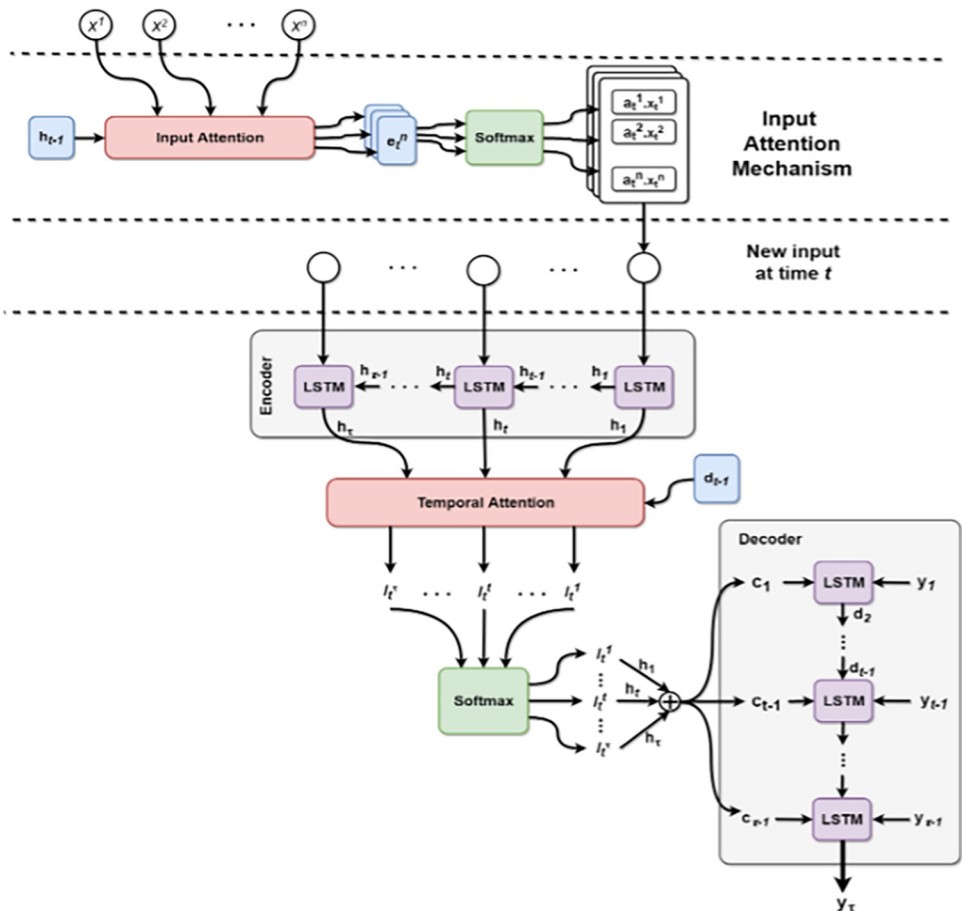

**Fig 6. Forecasted results.**

health risk factors directly cause all cervical cancers and 4.4% of prostate cancers and deaths. Risk factors cause 84.1 percent of tracheal, bronchus, and lung cancer, 71.8 percent of laryngeal cancer, 78.94 percent of lip and oral cavity, 69.8 percent of esophageal, and 64 percent of nasopharynx cancer cases and deaths. Health risk factors did not affect brain and nervous system cancer, pulmonary sarcoidosis, lymphatic filariasis, malignant skin melanoma, non-melanoma skin cancer, or testicular cancer. Health risk factors are linked to > 50% of 7 of 26 cancer types. Health risks cause 43.2% of colon and rectum cancers, 41.5 percent of liver cancer, 36.5 percent of uterine cancer, 42.3 percent of kidney cancer, 52 percent of bladder cancer, 27.5 percent of pancreatic cancer, 34.9% of breast cancer, 21.78% of stomach cancer, and 35% of gallbladder and biliary tract cancers. The bar's height and color indicate how much health risk factors contribute to cancer incidence and death. Risk factors cause 7.9% of thyroid cancer and 7.8% of Non-Lymphoma Hodgkin's incidence and mortality. The existing graph appears to have an issue with the presentation of the cervical cancer risk variables. The shade and bar height provide the wrong idea, even though they are meant to indicate the risk associated with obesity. The Centers for Disease Control and Prevention (CDC) reports that the Human Papillomavirus (HPV) is the causative agent of cervical cancer in around 90% of cases [151]. However, being overweight is not a primary cause. It isn't the main cause of the condition, but it could weaken the immune system and impact HPV progression in some circumstances. The existing layout of the graph, which uses color shading and bar height to portray the same data,

incorrectly suggests that obesity significantly increases the risk of cervical cancer [152,153]. This might be because of an error in the data mapping process or because the graph is lacking a data point that would indicate the risk of HPV infection. Having two separate data points for cervical cancer would be excellent for a clearer picture[154]. A large value, maybe the entire height of the bar or a dark shade, would indicate a high risk of HPV infection, whereas a little value, maybe a minimal height of the bar or a light shade, would indicate a low risk of obesity. It is essential to have labels that clearly explain the meaning of the colors and heights in order to prevent any misunderstandings. These changes will make the graph a better tool for learning about the real cervical cancer risk factors[155].

## 4.4 Forecasting results

All models were trained and tested 10 times in the same environment. We test on a GPU cluster of eight NVIDIA 1080Ti 32GB in simultaneously. Mini batching (128-person) was employed during training. A joint Stochastic Gradient Descent and Adam optimizer with 0.001 learning rate were used. We use 1000 training epochs and an early stopping strategy to cease when training losses do not decrease after 20 epochs. Our goal function was Mean Squared Error. Dropping 20% on all layers of each model we tested prevents overfitting. LSTM models are susceptible to overfitting, particularly when data is sparse. This can happen when the model grasps patterns that are unique to the training data but do not apply effectively to new data. Fig 6 presents the experimental results from the testing of cancer data on the Double Exponential Smoothing method. It is observed that the Double Exponential Smoothing model diverges as the years increase, even after testing data points of two years, unlike the neural networks models which undulate evenly and smoothly along the timeline [127,144,156]. This could be due to the absence of seasonality in the Double Exponential Smoothing, whereas the neural networks approach and factor the seasonality. After the model testing, it is realized that the Double Exponential Smoothing has an average accuracy of 88.50% in the prediction of cancer data.

This implies that the Double Exponential Smoothing method may help in short-term predictions (1–6 months), however, this method is comparatively low/weak in accuracy and not good for a long-term cancer projection for effective policy formulation. To test our model's network, we first conduct experiments on nine existing prediction state-of-the-art model networks, including: A Gated Recurrent Unit (GRU), A Bidirectional Gated Recurrent Unit (BiGRU), A Sequence-to-Sequence Gated Recurrent Unit (Seq-2-Seq GRU), A Bidirectional Sequence-to-Sequence Gated Recurrent Unit (Seq-2-Seq BiGRU), A Long short-term memory (LSTM), A Bidirectional Long short-term memory (BiLSTM), A Sequence-to-Sequence Long short-term memory (Seq-2-Seq LSTM), A Sequence-to-Sequence Bidirectional Long short-term memory (Seq-2-Seq BiLSTM), A Sequence-to-Sequence Long short-term memory with Input Attention (Seq-2-Seq LSTM + Input Attention). The average cancer data testing and predicting accuracy scores of existing models are compared [150,156]. The models underwent 10 rounds of training and evaluation in a consistent context. Each subsequent model replaced the training loss of the previous models. The initial testing findings of the Gated Recurrent Unit (GRU) model, as demonstrated in Fig 6, indicate an accuracy of 98.26%. Subsequently, the performance of the Bidirectional Gated Recurrent Unit model, consisting of 128 units, was evaluated, yielding an average accuracy of 97.1 percent. After the completion of the third round of testing, the Gated Recurrent Unit Sequence to Sequence network consisting of 128 units demonstrated an accuracy of 88.23%. The Bidirectional Sequence-to-Sequence Gated Recurrent Unit that was displayed was subjected to four tests. The Long short-term memory model, which incorporated post-training data training, had an average accuracy of 95.34

percent, whereas this model achieved 64.36 percent. Subsequently, the performance of the Bidirectional Long short-term memory Network, consisting of 128 units, was evaluated and yielded an average accuracy of 94.31 percent. Sequence-to-Sequence Long short-term memory, Bidirectional Long short-term memory, and long short-term memory with Input Attention were directly tested and achieved average accuracies of 89.21, 94.30, and 86.86. Finally, the set of testing is implemented on our proposed TSA-LSTM model. Just like the other State of the Art models, we trained ours and evaluated it ten times in the same environment. Each model overwrote the training loss of the previously implemented models. The result shows an average accuracy of 98.23%.

## 4.5 Quantitative analysis of models

Using real forecasts is crucial for accuracy evaluation. Thus, residual size does not predict genuine forecast errors. Only by testing a model on new data not utilized when fitting it can forecast accuracy be verified. Data is usually split into training and test data when picking models, with the former used to estimate forecasting technique parameters and the latter to assess accuracy. Since the test data is not used to forecast, it should give a good indicator of how well the model will forecast fresh data. In this study, training lasted from February 1990 to March 2021 and testing was from We also increased the test size to the maximum prediction horizon. We presume that a model that matches the training data well will not forecast well when creating our new model [157]. A model with enough parameters produces a perfect match every time. Overfitting a model is as harmful as not finding a pattern in data. Some publications call the test set the "hold-out set" since these data are "held out" of fitting data. This study uses "training data" and "test data" instead of "in-sample data" and "out-of-sample data" as references do.

We defined forecast error as the difference between an actual value and its forecast in this investigation. Thus, an error is the unpredictable component of an observation, not a mistake. Write it as:

$$e_{T+h} = y_{T+h} - \hat{y}_{T+h|T}, \quad \text{(Eq 19)}$$

Where the training data is given by $\{y_1, \ldots, y_T\}$ and the test data is given by $\{y_{T+1}, y_{T+2}, \ldots\}$.

Forecast errors differ from residuals in two ways. Remainders are calculated on the training set and forecast mistakes on the test set. Second, residuals use one-step forecasts whereas forecast errors use multi-step forecasts. We use scaled-dependent errors to quantify forecast accuracy. This study's forecast mistakes are on the same scale as the data hence the models' accuracy metrics are scaled-dependent as the cancer data is in numerous time series with the same units (date and cancer incidence). Different forecasting models are compared by their error and accuracy rates. Forecasting models vary in accuracy. Table 2 compares double exponential smoothing, deep learning techniques, and our model. As mentioned earlier, accuracy is a key indicator of predictive algorithm success [158].

Our algorithm has the maximum accuracy of 98.96 percent and the lowest error rate of 3.24 percent. Double exponential smoothing predicts cancer incidence 9th among deep learning methods. We employed one double exponential smoothing approach, but the alpha-beta values with the fewest mistakes were used during implementation. Thus, the amount of double exponential methods utilized will not affect performance but may decrease it. Again, deep learning systems outperform double exponential smoothing in cancer data prediction accuracy. Comparatively, our model is the most accurate. Our algorithm uses two-stage attention. Attention models might provide interpretability while keeping deep learning's flexibility and versatility. Comprehending the prediction process of the model might be challenging because

**Table 2. Summary of quantitative comparison of models.**

| Models | Average Score | Position | Percentage of Error |
| --- | --- | --- | --- |
| **TSA-LSTM NETWORK** | **98.96** | **1st** | **3.24%** |
| LSTM SEQ-2-SEQ NETWORK WITH INPUT ATTENTION | 96.97 | 2nd | 5.03 |
| LSTM NETWORK | 93.21 | 3rd | 4.92 |
| GRU | 96.37 | 4th | 5.94 |
| LSTM SEQ-2-SEQ NETWORK | 91.43 | 5th | 8.51 |
| BI-GRU | 94.23 | 6th | 8.87 |
| BILSTM SEQ-2-SEQ NETWORK | 96,56 | 7th | 8.43 |
| BILSTM NETWORK | 93.43 | 8th | 8.53 |
| DOUBLE EXPONENTIAL TECHNIQUE | 87.76 | 9th | 10.54 |
| GRU SEQ-2-SEQ NETWORK | 76.87 | 10th | 31.56 |
| BI-GRU SEQ-2-SEQ NETWORK | 61.43 | 11th | 68.64 |

of the intricate structure of LSTMs. In cancer research, the inability to grasp the model's reasoning can be a drawback for creating novel therapy approaches. We initially use an input attention technique to adaptively extract important driving series at each time step by referencing the encoder's hidden state. We pick relevant encoder hidden states across all time steps in the second stage using temporal attention. Our model can predict and interpret well with this dual-stage attention method [123,159,160]. Time series prediction and feature learning in computer vision tasks are possible with our model. This implies that our model forecasts cancer data better than others. Our algorithm may have a low margin error since the model's attentions provide interpretability and flexibility.

## 4.6 Qualitative algorithm description

This section describes each algorithm based on three essential time series observations on our predicting outcomes. 1). predict smoothness 2) movement distance and orientation; 3) training average time. These observations may help explain algorithm behavior on cancer data. We discuss the neural network and DES algorithm training procedures to elucidate the observations. The first section gives four rudimentary forecasts and the second four refined ones. Primitive forecasts are training's first projections without error margin. Refined forecasts are error margin-influenced. Both crude and sophisticated forecasts improve accuracy. The last two training instances just increased the error margin [161,162]. The two training instances and error margin are meant to help refined forecast training instances match the genuine trend [163]. The predicting results are supported by the error margin and two training instances using three fundamental time series data. The training procedures are exclusive to DES. However, as DES is not a neural network, the training instance only constituted a piece of the process. We assessed the effectiveness of each method in improving forecast accuracy by applying filtering techniques. Smoothing helps to clarify the trends in forecasting [35,164] Smoothing describes time series trend components. Smoothing can be linear, undulating, or non-undulating [165]. The optimal neural network algorithm increases undulating trends and minimizes non-undulating.

According to Table 3, the best neural network has a whole undulating pattern for forecast results and full distances between forecast results and true trends. Such a neural network needs to be trained for over 1700 seconds. Several experiments led to this conclusion. We trained our data with basic LSTM and GRU. We found that the simple LSTM model outperforms the GRU. This supports our decision to build on the basic LSTM with an attention network [166]. We fed the basic LSTM algorithm a one-stage attention network to improve performance. Our

**Table 3. Observations that influence the accuracy of the algorithm.**

| Algorithm Name | Filtering (Partial / Full) | Distance (Partial/full) | Average Time of Training (Seconds) |
|---|---|---|---|
| GRU | Full non-undulating | Partial | 522 |
| BiGRU | Partial non-undulating | Partial | 867 |
| Seq-2-Seq GRU | Partial non-undulating | Partial | 945 |
| Seq-2-Seq BiGRU | Partial non-undulating | Partial | 1245 |
| LSTM | Full undulating | Full | 654 |
| BiLSTM | Full undulating | Partial | 876 |
| Seq-2-Seq LSTM | Partial non-undulating | Partial | 865 |
| Seq-2-Seq BiLSTM | Partial non-undulating | Full | 1685 |
| Double Exponential Technique | Linear | Full | - |
| Seq-2-Seq LSTM + Input Attention) | Full non-undulating | Partial | 1433 |
| TSA-LSTM | Full undulating | Full | 1865 |

next hypothesis is that a two-stage attention network as an input to the basic LSTM would improve performance. We found that the two-stage attention network improves cancer data performance but requires many hours of training.

## 4.7 Prediction of cancer incidences using the TSA-LSTM model

Fig 6 shows our unique TSA-LSTM model's experimental cancer incidence forecasts. Predictions for new cancer cases are until 2025. The new cancer incidence estimates are increasing linearly, indicating high cancer rates until 2025. Our actual numbers ending 2021 suggest cancer cases may rise 9% by December 2020 and 18% by December 2021. Additionally, 2022 and 2023 are predicted to see 18% and 21% more cancer cases. Cancer incidences are expected to grow by 20% and 22% in 2024 and 2025 [167,168]. The forecasts are reasonable and incremental, with close prediction intervals reflecting previous data fluctuation. Table 4 summarizes the total projected cancer incidence in December each year. Additionally, it has been determined that the eating of particular foods is likely to elevate the risk of developing cancer. For example, consuming meals that induce hyperglycemia has been associated with an elevated risk of developing colorectal, stomach, and breast malignancies [31,62]. High-temperature cooking techniques, including grilling, frying, and barbecuing, can generate heterocyclic amines (HA), which have been linked to the development of cancer and various other ailments [62]. A high dairy intake is associated with an increased risk of getting prostate cancer. According to [169] an increased consumption of whole milk significantly heightens the likelihood of illness progression and mortality. Furthermore, it is widely acknowledged that physical inactivity is the leading cause of breast, colon, and corpus/uterine malignancies among women in the United States. Our projection shows rising US cancer rates by 2025. As shown in sections 4–20,

**Table 4. Summary of predicted cancer incidence rate.**

| YEAR | CANCER PREDICTION RATE |
|---|---|
| 2020 | 10% |
| 2021 | 13% |
| 2022 | 16% |
| 2023 | 18% |
| 2024 | 20% |
| 2025 | 22% |

lifestyle decisions and other risk factors contribute to the rising cancer rate [170]. Lifestyle choices including alcohol, smoke, inactivity, and poor diet have boosted obesity in the US, and this trend is expected to continue. Previous studies have connected poor lifestyle choices to cancer. The International Agency for Research on Cancer showed increased colon, breast, and endometrial cancer risk. Smoking causes breast and lung cancer, research shows [24,25,171].

Our results reveal that the Double Exponential Smoothing model diverges on cancer data over time. but bad for long-term forecasting. One of our 11 study algorithms. Comparatively, our model has a maximum accuracy of 98.96 percent. LSTM model sequenced by our technique used a two-stage attention sequence. We forecast US cancer incidence growth using our two-stage attention model with 2021 as our baseline (Table 4). Furthermore, our research findings suggest that the Double Exponential Smoothing model exhibits a divergence from the observed outcomes over time. This result is attributable to the Double Exponential Smoothing model's absence of seasonal variation. This indicates that while the Double Exponential Smoothing method may be advantageous for short-term forecasts (1–6 months), it is not considered suitable for long-term forecasting. One of eleven algorithms that were implemented in our research. In comparison to alternative models, our proposed model exhibits the maximum magnitude of accuracy at 97.76 percent. The attention sequence to sequence LSTM (TSA-LSTM) model was utilized by our method in two stages. The rate of growth in cancer incidence in the United States is predicted using our two-stage attention model, with 2017 serving as the baseline year. By December of 2020, 2021, 2022, 2023, 2024, and 2025, the incidence of cancer is projected to grow by 9 percent, 12 percent, 14 percent, 16 percent, 19 percent, and 21 percent, respectively.

## 5. Discussion

In this study the exponential smoothing (ES) method can be considered as weighted moving averages after Brown (1959,1962) and Holt (1960) studied stock control system forecasting, it was introduced. This is the weighted average of historical observations [172]. Past observations are weighted exponentially decreasingly, while recent observations are weighted most heavily. Since it adapts well to short-term forecasting, it makes good daily predictions. SES, the simplest exponential smoothing approach, forecasts data without trend or seasonality [173]. SES is decreasing past observation weights exponentially. SES estimates future values using the preceding period's forecast and forecast error. The Holt-Winters smoothing approach emphasizes seasonality or trend, unlike most ES methods. Holt (1957) and Winters (1960) introduced smoothing factors, trend (a slope), and seasonality (cyclical, recurring pattern) to this method to capture seasonality and trend. Triple exponential smoothing comprises the value, trend, and seasonality of the time series. The Holt-Winters' adaptive model allows the level, t, rend, and seasonality patterns to change corresponding to time [171]. Past observations are getting exponentially less SES weights. The forecast for the preceding period and the forecast error are used to estimate future values in SES. Holt-Winters smoothing considers seasonality or trend, unlike most ES methods. By incorporating smoothing variables, trend (a slope), and seasonality (cyclical, recurring pattern), Holt (1957) and Winters (1960) improved this method to incorporate seasonality and trend. Time series value, trend, and seasonality are called triple exponential smoothing. Recent observations are weighted greater in exponential smoothing. These models extrapolate data trends and seasonality [174]. Exponential smoothing iteratively forecasts from future values of a time series form weighted averages of past values [175]. The use of the TSA-LSTM model in clinical settings is accompanied by several specific limitations that must be surmounted in order to fully exploit its capabilities. Error-prone and fragmented clinical data and the need for large amounts of labeled data for efficient training may reduce

model precision and dependability. Healthcare professionals have difficulty deciphering the model as a result of its complexity. To foster confidence and promote adoption, offer straightforward explanations for predictions and showcase the value of the features being proposed. Furthermore, healthcare organizations that lack expertise and infrastructure in machine learning may encounter difficulties in training and deploying deep learning models such as TSA-LSTM as a result of the computational resource demands[176]. To ensure the model's ethical clinical deployment, patient privacy, permission, bias mitigation, and healthcare data rules must be carefully considered. The TSA-LSTM model must also be smoothly linked into clinical procedures and electronic health record systems for practical adoption and user acceptance. This highlights the need for data scientists, healthcare providers, legislators, and regulators to work together to address these problems and use predictive analytics in healthcare.

The deep learning algorithms have outperformed computer vision in recent years, but their time series models are still shallow. Many data scientists in most industries use autoregressive models instead of deep learning. Interpretability, restricted data, ease of use, and training cost are common reasons for choosing these methods [177]. There is no single solution to these difficulties, but deep models with care make a strong case. They often increase performance (other vanilla LSTMs/RNNs) and interpretability. In many circumstances, they are faster than RNN/LSTM. Attention networks are effective for embedding categorical inference in deep neural networks [178]. Brain networks assist in awareness, sensory orientation, and conflict resolution. In most sequence transduction models, large recurrent or convolutional neural networks encode and decode locally without global information. Cancer and predictive healthcare analytics demonstrate the TSA-LSTM model's utility and limits. The algorithm predicts cancer using historical data, demographics, environment, and genetic markers. It can assist governments and healthcare practitioners find patterns and cure cancer. It predicts cancer and chronic illness mortality over time using patient data and risk indicators, which may help clinicians improve therapy. This algorithm predicts therapeutic reactions using clinical data and illness development, helping clinicians optimize treatment regimens and select the best drugs. This tool analyzes patient data for temporal trends to diagnose cancer and other diseases early. This technological advancement empowers medical professionals to proactively enhance treatment outcomes through the identification of gradual fluctuations in health indicators that could potentially signify the start of an illness. Healthcare systems can use the TSA-LSTM model to predict future healthcare needs, such as cancer case volume, and allocate resources strategically. Oncology departments and facilities may improve resource use, capacity planning, and healthcare delivery. In general, the TSA-LSTM model improves prognostics, personalized medicines, and evidence-based decision-making in oncology and other healthcare sectors. The utilization of the model's sophisticated temporal relationships and sequencing functions has the potential to enhance health outcomes, optimize resource allocation, and improve patient care. Critical healthcare applications exist for TSA-LSTM. The prediction of early cancer diagnosis, therapy responses, and cancer incidence and mortality. Beyond the realm of oncology, medical practitioners have the ability to optimize resource allocation, customize patient treatment, and enhance overall patient outcomes by utilizing the model's comprehensive temporal sequences and patterns. This limitation led to the development of attention networks to capture long-distance interactions. Attention networks are increasingly utilized due to its fully parallelized computation and flexibility in modeling dependencies [179]. According to [180]. The best models use attention to link the encoder and decoder. Recurrent and convolutional models based only on attention mechanisms perform better, are more parallelizable, and take less time to train on two machine translation tasks. The work of (Ramachandran et al., 2019 [181]; and Sand uddle & Bashir, 2022 [182]) A simple ResNet model that replaced all spatial convolutions with self-attention showed attention networks'

impact. This fully self-attentional model surpassed the baseline with less categorization parameters. Later investigations on attention in generic deep neural networks show ongoing categorization and detection with different models.

Based on qualitative algorithm description, neural network techniques should have equally undulating or non-undulating forecasting instances in both training parts. We describe full or partial undulating and non-undulating filtering. We have two fully undulating training instances with equal tendencies. Fully non-undulating occurs when both training instances are non-undulating. Partial means undulating and non-undulating training instances are not equal [35,136]. In contrast, the DES trend is largely linear. Distance describes the intervals and slopes between the underlying trend and each anticipated result [183]. A neural network method functions best when primitive and refined training cases predict the true trend. Additionally, a neural network performs best when anticipated slopes match true trends. We call these lengths full or partial[184]. When both training instances give forecasting results closer to genuine trends, full distance is reached. Also, partial distance occurs when only one training instance forecasts trends closer to reality. We observe that the simple LSTM model has a full distance for most anticipated results since the underlying trend is in the middle. However, BILSTM outcomes have a partial distance because one anticipated result is far from the true trend [185]. Based on the TSA-LSTM Model connection between alcohol taking and cancers have been found. It has been found that cancer deaths attributable to alcohol consumption are 3.5% in the USA [186]. The consumption of certain foods has also been found to likely increase the probability of developing cancer. For instance, foods that cause blood glucose levels to rise are connected with an increased risk of breast, stomach, and colorectal cancers Moreover, Physical inactivity is seen as a principal cause of female breast, colon, and corpus/uterine cancers in the USA [68,186].

## 6. Conclusion and policy implication

There are various subtypes of cancer. Cancer ranks second in global prevalence, accounting for one in every six fatalities. The incidence of cancer is projected to increase as a result of factors such as lifestyle changes, population growth, and extended life expectancy. Thus, an accurate cancer forecast will enable health professionals, government agencies, and the public to prepare for its detrimental repercussions. The search for a good forecasting tool has been difficult for stakeholders. A solid forecasting approach offers accurate and understandable outcomes, not complex methodology. Current research divides forecasting approaches into three categories: artificial intelligence, statistical, and hybrid. Recent interest in statistical models has focused on Exponential Smoothing (ES) and (DES). ES, DES are weighted moving averages. Tableau, a popular data visualization and forecasting application, uses exponential smoothing. These approaches can reliably anticipate short-term data. Second, deep learning/machine learning algorithm forecasting techniques use intelligence and flexibility to improve prediction models. In deep learning, Artificial Neural Network (ANN) models are optimal for real-time estimations. The two are combined in the third algorithm. The health sector has employed various algorithms in pursuit of research objectives; nevertheless, the optimal algorithm for ensuring the accuracy of cancer data has yet to be investigated. This research endeavors to identify the most precise cancer data prediction software. The study does this by Testing double exponential smoothing on cancer data and comparing it to other deep-learning techniques. Second, creating a cancer data-accurate deep learning algorithm. Third, predict cancer incidence rise by 2025 using our innovative approach.

We contributed three parts to the study. First, Cancer Prediction using Double Exponential Smoothing and Deep Learning Algorithms Our results reveal that the Double Exponential

Smoothing model diverges on cancer data over time. This is because Double Exponential Smoothing lacks seasonality. Thus, Double Exponential Smoothing may be useful for short-term predictions (1–6 months) but not long-term predictions. This study examined 11 methods, and double exponential smoothing predicted cancer incidence 9th best. Thus, DES has poor long-term cancer prognosis and data accuracy. Double Exponential Smoothing is unsuitable for long-term cancer projection in policy planning. Our model has a maximum accuracy of 98.96%. Our two-stage LSTM model sequencing method. While maintaining deep learning's flexibility and versatility, attention models may give interpretability. We reference the encoder's hidden state to adaptively extract significant driving series at each time step using input attention. We pick relevant encoder hidden states across all time steps in the second stage using temporal attention. Our model can predict and interpret well with this dual-stage attention method. Time series prediction and feature learning in computer vision tasks are possible with our model. This implies that our model forecasts cancer data better than others. Our approach may have great accuracy since the model's attentions provide interpretability and flexibility. Third, 2025 cancer incidence forecast Using 2021 as a baseline, our innovative two-stage attention model predicts cancer incidence growth. The stipulated increase in cancer incidences is 14% by December 2020 and 12% by December 2021. Additionally, 2022 and 2023 are predicted to have 14% and 16% more cancer cases. Cancer incidences are expected to grow by 19% and 21% in 2024 and 2025. Lifestyle decisions including alcohol drinking, tobacco smoking, physical inactivity, and poor eating practices that lead to obesity in the US are projected to raise cancer rates. We advocate (i) widespread safety education on the harmful impacts of unhealthy lifestyle choices at home at work to lower cancer rates in the US. (ii) Lifestyle and cancer research should receive government financing. (iii). Provincial governments should also try new approaches to discourage unhealthy lifestyles. Reforms should limit the amount of tobacco or alcohol sold per day.

According to future prediction, we created a novel two-stage attention model technique that can forecast cancer incidence monthly with better accuracy than existing algorithms. Next, we examine engineering, health, economy, and social research directions. The engineering community can improve cancer prediction. Researchers can use our model as a baseline because long-term prediction demands great accuracy. Our unique approach has 98.96 percent accuracy. Next steps should boost accuracy and shorten training. This study is ongoing; therefore, we hope to build a model that can detect cancer cases weekly and daily instead of monthly. This strategy will emphasize cancer case management—early detection and death. We hope to achieve this through network optimization. Increased cancer cases may impact economic indicators. More cancer cases equal more treatment costs. This represents a national fiscal burden because cancer treatment requires large amounts. Cancer rise may affect GDP and inflation in future studies. Society views cancer survivors as having psychological and emotional disorders following treatment. This hypothesis can be tested by future research.

## Supporting information

**S1 Appendix. Centers for Disease Control and prevention (CDC).**
(DOCX)

**S2 Appendix. Linear, quadratic, or cubic data.**
(DOCX)

**S3 Appendix. TSA-LSTM model.**
(PNG)

**S1 Dataset.**
(XLSX)

## Acknowledgments

The author is profoundly grateful to the instructors for their encouragement and guidance, and will respond to their insightful advice. Fuzhou University of Technology's educational opportunities are sincerely acknowledged. China's National Social Science Foundation (22BGL007). GY-H-24176: Research on marine environmental protection policies and low-carbon ship emission control. The "Research on the Impact of Low Carbon Strategies on the Rural Revitalization Strategy in Fujian Province" (GY-S20014) project has been initiated by Fujian University of Technology.

## Author Contributions

**Conceptualization:** Rabnawaz Khan.

**Data curation:** Rabnawaz Khan, Wang Jie.

**Formal analysis:** Rabnawaz Khan.

**Funding acquisition:** Rabnawaz Khan.

**Investigation:** Rabnawaz Khan, Wang Jie.

**Methodology:** Rabnawaz Khan.

**Project administration:** Rabnawaz Khan.

**Resources:** Wang Jie.

**Software:** Rabnawaz Khan, Wang Jie.

**Supervision:** Rabnawaz Khan.

**Validation:** Rabnawaz Khan.

**Visualization:** Rabnawaz Khan.

**Writing – original draft:** Rabnawaz Khan, Wang Jie.

**Writing – review & editing:** Wang Jie.

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
