## [Decision Letter · Decision Letter 0]

29 Jan 2024

PONE-D-23-37813Using the TSA-LSTM Two-Stage Model to Predict Cancer Incidence and MortalityPLOS ONE

Dear Dr. Khan,

Thank you for submitting your manuscript to PLOS ONE. After careful consideration, we feel that it has merit but does not fully meet PLOS ONE’s publication criteria as it currently stands. Therefore, we invite you to submit a revised version of the manuscript that addresses the points raised during the review process.

We look forward to receiving your revised manuscript.

Kind regards,

Jayesh Soni

Academic Editor

PLOS ONE

Journal Requirements:

2. For studies reporting research involving human participants, PLOS ONE requires authors to confirm that this specific study was reviewed and approved by an institutional review board (ethics committee) before the study began. Please provide the specific name of the ethics committee/IRB that approved your study, or explain why you did not seek approval in this case.

4. Please note that PLOS ONE has specific guidelines on code sharing for submissions in which author-generated code underpins the findings in the manuscript. In these cases, all author-generated code must be made available without restrictions upon publication of the work. Please review our guidelines at https://journals.plos.org/plosone/s/materials-and-software-sharing#loc-sharing-code and ensure that your code is shared in a way that follows best practice and facilitates reproducibility and reuse.

The author responds to insightful advice and is extremely appreciative of the instructors for all the encouragement and guidance they have given. The educational opportunities provided by the Fujian University of Technology are gratefully acknowledged. National Social Science Foundation of China (22BGL007), Fujian Zhi-lian-yun Supply Chain Technology and Economy Integration Service Platform from the Fujian Association for Science and Technology, the Fujian-Kenya Silk Road Cloud Joint R&D Center (2021D021) from Fujian Provincial Department of Science and Technology, the Fujian Social Sciences Federation Planning Project (FJ2021Z006), and General program of Fujian Natural Science Foundation (2022J01941). Fujian University of Technology Launch Project “Research on the Impact of Low Carbon Strategies on the Rural Revitalization Strategy in Fujian Province” (GY-S20014). He is grateful to his loved ones for their support as he pursues his goal of earning a doctorate. He cannot ignore the value of the disabled student who, in the name of success, is willing to sacrifice anything, including their feelings, in order to succeed. He knows that love is the one that holds him up and gives him the energy to keep going. 

Additional Editor Comments:

Having carefully gone through the manuscript, I find the subject matter both relevant and timely. Your approach to using the TSA-LSTM model for predicting cancer incidence and mortality is innovative and could potentially contribute significantly to the field. However, I have a few questions that I believe, once addressed, could further strengthen the paper:

Model Validation and Comparison: Could you provide more details on how the TSA-LSTM model's predictive performance compares to existing models? Specifically, it would be beneficial to see comparisons to other prevalent models in cancer incidence and mortality prediction, focusing on accuracy, reliability, and computational efficiency.

Data Preprocessing and Feature Selection: Your paper mentions data preprocessing and sample expansion design. Could you elaborate more on the feature selection process for the TSA-LSTM model? How were the variables chosen, and what was the rationale behind including or excluding certain variables?

Interpretability of Model Results: The TSA-LSTM model seems to be a complex structure. How do you address the challenge of interpretability of the model's results, especially for healthcare professionals who might not have a deep understanding of machine learning models?

Practical Applications and Limitations: Finally, could you discuss the practical applications of your model in real-world healthcare settings? Are there any specific limitations or challenges in implementing this model in a clinical environment?

I believe that addressing these and reviewers' queries will not only clarify certain aspects of your work but also provide a more comprehensive understanding for readers and practitioners in the field. I look forward to your responses and am excited about the potential impact of your research in the domain of cancer prediction and analysis.

Reviewers' comments:

Reviewer's Responses to Questions

**Comments to the Author**

1. Is the manuscript technically sound, and do the data support the conclusions?

Reviewer #1: Yes

Reviewer #2: Yes

2. Has the statistical analysis been performed appropriately and rigorously? 

Reviewer #1: Yes

Reviewer #2: Yes

3. Have the authors made all data underlying the findings in their manuscript fully available?

Reviewer #1: Yes

Reviewer #2: Yes

4. Is the manuscript presented in an intelligible fashion and written in standard English?

Reviewer #1: Yes

Reviewer #2: Yes

5. Review Comments to the Author

Reviewer #1: The research is interesting, however, this paper is suggested to be published in PLOS ONE after authors have addressed the comments listed below:

1. For the literature review, authors should refer to more recent research. The current references are not up to date. Authors need to refer to more ISI/Scopus research work instead of the online/conference resources.

2. The literature review need to be further improve. The current review is too general and lengthy. The literature review need to be more specific and need to be tie back to the domain of the research. Please provide a summary at the end of the section to summary the research insights.

3. Authors need to identify the research gap and highlight the contribution(s) of the research work in the manuscript to shows the novelty of the research. The proposed methodology existed in the literature, please highlight the novelty of the research work in the manuscript.

4. Authors should compare the proposed techniques with the state-of-arts to further prove the contribution(s)/novelty of the research work.

5. More scientific reasoning should be added in the experimental results' explanations.

Reviewer #2: The literature review was thorough and covered all the background information needed to understand the work in this paper.

The results and the subsequent discussion were interesting, specially with the different deep learning algorithms and their performance on the dataset.

There were a few minor grammatical errors, but nothing major.

6. PLOS authors have the option to publish the peer review history of their article (what does this mean?). If published, this will include your full peer review and any attached files.

Reviewer #1: No

Reviewer #2: No

---

## [Author Response · Author response to Decision Letter 0]

13 Feb 2024

Response of editor and reviewers’ response is already attached

---

## [Decision Letter · Decision Letter 1]

24 Mar 2024

PONE-D-23-37813R1Using the TSA-LSTM Two-Stage Model to Predict Cancer Incidence and MortalityPLOS ONE

Dear Dr. Khan,

Thank you for submitting your manuscript to PLOS ONE. After careful consideration, we feel that it has merit but does not fully meet PLOS ONE’s publication criteria as it currently stands. Therefore, we invite you to submit a revised version of the manuscript that addresses the points raised during the review process.

We look forward to receiving your revised manuscript.

Kind regards,

Jayesh Soni

Academic Editor

PLOS ONE

Additional Editor Comments:

The paper claims a long-term projection accuracy of 98.96%. How was this accuracy validated, especially considering the complexity of cancer incidence and mortality trends over time?

The manuscript discusses using cubic spline interpolation to expand data points from 32 to 420 for more detailed analysis. How does this significant increase in data points affect the model's performance and generalizability, especially in terms of overfitting or underfitting?

Reviewers' comments:

Reviewer's Responses to Questions

**Comments to the Author**

1. If the authors have adequately addressed your comments raised in a previous round of review and you feel that this manuscript is now acceptable for publication, you may indicate that here to bypass the “Comments to the Author” section, enter your conflict of interest statement in the “Confidential to Editor” section, and submit your "Accept" recommendation.

Reviewer #3: All comments have been addressed

Reviewer #4: All comments have been addressed

2. Is the manuscript technically sound, and do the data support the conclusions?

Reviewer #3: Yes

Reviewer #4: Yes

3. Has the statistical analysis been performed appropriately and rigorously? 

Reviewer #3: Yes

Reviewer #4: Yes

4. Have the authors made all data underlying the findings in their manuscript fully available?

Reviewer #3: Yes

Reviewer #4: Yes

5. Is the manuscript presented in an intelligible fashion and written in standard English?

Reviewer #3: Yes

Reviewer #4: Yes

6. Review Comments to the Author

Reviewer #3: (No Response)

Reviewer #4: 1.The cancer data are known to be complex and heterogeneous. How well does the TSA-LSTM Two-Stage Model handle such complexities and heterogeneity?

2.Does the methodology have any limitations or have it been made considering of some assumptions that could possibly lessen the accuracy and/or generalizability of the outputs?

3. Have you ever been to a live concert of your favorite band playing your favorite songs and experienced the sound coming out of state-of-the-art sound systems that lasts your entire drive home?

4.Where did the cancer data come from and how were the process and the quality controlling assurance implemented to ensure the data was correct and reliable?

5.How did the statistical analysis specifically indicate the missing, unreliable or unpleasant data?

6.Was there some difficulty united the heterogeneous datasets with varying origins?

7.In what way the predictive precision of the two-stage model of TSA-LSTM is better than other models which are called the state-of-the-art of cancer prediction.

8.Was there suitable evaluation metrics used with the models' output, e. g. sensitivity, specificity, ROC or precision -recall curves?

9.Was there any data related difference in treatment of cancer through different cancer types or stratum?

10.Does the TSA-LSTM Two-Stage Model model depict the crucial factors and can provide information about the trends in cancer incidence and mortality rates?

11.How without difficulty understandable are the predictions generated by the version for healthcare practitioners and policymakers?

12. Were efforts made to interpret the learned representations or function significance inside the model?

Thirteen.

13.What capacity clinical programs should get up from the predictions generated through the TSA-LSTM Two-Stage Model?

14.Has the version been validated in medical settings, and in that case, what have been the consequences?

15. What are the results of the model's predictions for healthcare resource allocation, prevention strategies, or remedy planning?

16.Were there any moral considerations addressed in the development or deployment of the TSA-LSTM Two-Stage Model, along with privacy issues or bias mitigation?

17.How would possibly the usage of predictive fashions in most cancers prediction effect healthcare disparities or get admission to to care?

18. Are there any accidental outcomes or dangers related to counting on algorithmic predictions for healthcare choice-making?

7. PLOS authors have the option to publish the peer review history of their article (what does this mean?). If published, this will include your full peer review and any attached files.

Reviewer #3: No

Reviewer #4: **Yes: **K.R.Sekar

---

## [Author Response · Author response to Decision Letter 1]

1 Apr 2024

Response of review have been attached with date.

---

## [Decision Letter · Decision Letter 2]

22 May 2024

PONE-D-23-37813R2Using the TSA-LSTM Two-Stage Model to Predict Cancer Incidence and MortalityPLOS ONE

Dear Dr. Khan,

Thank you for submitting your manuscript to PLOS ONE. After careful consideration, we feel that it has merit but does not fully meet PLOS ONE’s publication criteria as it currently stands. Therefore, we invite you to submit a revised version of the manuscript that addresses the points raised during the review process.

We look forward to receiving your revised manuscript.

Kind regards,

Jayesh Soni

Academic Editor

PLOS ONE

Reviewers' comments:

Reviewer's Responses to Questions

**Comments to the Author**

1. If the authors have adequately addressed your comments raised in a previous round of review and you feel that this manuscript is now acceptable for publication, you may indicate that here to bypass the “Comments to the Author” section, enter your conflict of interest statement in the “Confidential to Editor” section, and submit your "Accept" recommendation.

Reviewer #3: (No Response)

Reviewer #5: (No Response)

2. Is the manuscript technically sound, and do the data support the conclusions?

Reviewer #3: Partly

Reviewer #5: No

3. Has the statistical analysis been performed appropriately and rigorously? 

Reviewer #3: N/A

Reviewer #5: No

4. Have the authors made all data underlying the findings in their manuscript fully available?

Reviewer #3: No

Reviewer #5: Yes

5. Is the manuscript presented in an intelligible fashion and written in standard English?

Reviewer #3: Yes

Reviewer #5: No

6. Review Comments to the Author

Reviewer #3: This paper proposed a TSA-LSTM two-stage model to predict cancer incidence and mortality. The idea is interesting and the research is complete. Some improvements should be made before the paper is accepted.

1. The length of the paper should be limited. The author needs to condense the paper and emphasize the key content.

2. The content of Appendix 4 and 5 should be exchanged. Besides, both of them are crucial components of the paper, which should be shown in the main body of the paper.

3. The results are not rigorous enough since the average score is not convincing. As a prediction method, RMSE and R2 should be listed to evaluate the performance. Besides, boxplots of errors are also suggested to be provided.

Reviewer #5: I am sorry to say that I have no idea what the authors intended or did in this long text. For example, the introduction mentions predicting disease outcomes, as "However, surgeons struggle most with illness outcome prediction," the data used are not data from which illness outcomes can be predicted, and no results or discussion of illness outcome prediction are given. The section titled as "review," it should be based on a systematic literature review process, or at least the literature search method should be described in the method section.

I believe that the authors aimed to develop a model used deep learning for projections of cancer incidence. I believe that the authors should focus on the aim and results.

p.3-6 Introduction

What the authors are doing is prediction of the future incidence of cancer. I suggest that the authors focus on that and describe which aspects of the methods currently used to cancer incidence prediction need to be improved, and the advantages of the authors' new methods.

p.7-13 2. Literature Review

What is presented in this section is hardly to title as review, so it would be appropriate to delete repetition and shorten it to be an introduction. If you want to make it as a review, you should either conduct a systematic review or describe how you selected the papers to be reviewed in the method section.

2.2 Description of Tableau

I do not understand the intent of describing Tableau. The authors do not use Tableau in their cancer incidence and mortality prediction model (Results of prediction for mortality were not shown in this draft.though.). Tableau seems to be used only to find factors to be included in the model or to visualize the results, there is no those figure or table though. If so, it is necessary to describe how the use of Tableau improves the search of risk factor and visualization of the result. For the usage in current draft, a sentence "Tableau was used to create the graph" in the method section is sufficient.

p.13-17, 3. Empirical Methodology

The dataset came from the Centers for Disease Control and prevention (CDC) and the American Cancer Society (Appendix 1).

Not only the description of the data were obtained from the Centers for Disease Control and prevention (CDC) and the American Cancer Society but also more information, for example, is it based on a registry system, what year to year incidence rates, when was the information on risk factors collected, for what populations, and by what methods?, etc., should be described. Brief description and citation of the article or website where they are described is possible. It is very important information what data were used for the model building and validation. This information help the readers to understand the results the authors obtained.

Figure 2.

There seems to be a lack of understanding of what the diagram created in Tableau shows; in Text, this diagram is used to show how bar charts can be easily created by drag-and-drop in Tableau, but for that purpose, it would be appropriate to show the Tableau startup screen and the operating procedure, I do not believe that it is the authors' aim to explain how to use Tableau, so I think that this diagram is NOT necessary.

Figure 3.

The y-axis is the incidence rate, and the notation of the x-axis and the legend are the same. This figure could be lead misunderstanding of readers that Tableau does not support such an easy adjustment. There is also a discrepancy between the figure and the text.

Figure 4.

Figure 4 seems shows the attributable risk of obesity to the incidence rate for each cancer. It is natural that breast cancer, which has a high incidence rate, would be larger, and the graph does not show what the authors intended. The wording also needs to be considered, since being related alone does not mean being the cause. It is a basic requirement of scientific papers that the appropriate units must be indicated on the graphs, so please do not forget to do so.

Figure 5.

The color shading and the height of the bars represent the same thing. It is well known that ninety percent of cervical cancers are caused by HPV (https://www.cdc.gov/hpv/parents/cancer.html), and 100% of cervical cancers are never caused by obesity.

7. PLOS authors have the option to publish the peer review history of their article (what does this mean?). If published, this will include your full peer review and any attached files.

Reviewer #3: No

Reviewer #5: No

---

## [Author Response · Author response to Decision Letter 2]

30 May 2024

Response of reviewers have been attached

---

## [Decision Letter · Decision Letter 3]

23 Oct 2024

PONE-D-23-37813R3Using the TSA-LSTM Two-Stage Model to Predict Cancer Incidence and MortalityPLOS ONE

Dear Dr. Khan,

Thank you for submitting your manuscript to PLOS ONE. After careful consideration, we feel that it has merit but does not fully meet PLOS ONE’s publication criteria as it currently stands. Therefore, we invite you to submit a revised version of the manuscript that addresses the points raised during the review process.

We look forward to receiving your revised manuscript.

Kind regards,

Amgad Muneer

Academic Editor

PLOS ONE

**Additional Editor Comments:**

Your paper has some issues with the figure's quality. The figures are blurry and not readable. Figure 2 Visual on creating a visualization in Tableau, the label seems not complete.  And the proposed TSA-LSTM design is blurry and not readable at all.

Reviewers' comments:

Reviewer's Responses to Questions

**Comments to the Author**

1. If the authors have adequately addressed your comments raised in a previous round of review and you feel that this manuscript is now acceptable for publication, you may indicate that here to bypass the “Comments to the Author” section, enter your conflict of interest statement in the “Confidential to Editor” section, and submit your "Accept" recommendation.

Reviewer #3: All comments have been addressed

2. Is the manuscript technically sound, and do the data support the conclusions?

Reviewer #3: Yes

3. Has the statistical analysis been performed appropriately and rigorously? 

Reviewer #3: Yes

4. Have the authors made all data underlying the findings in their manuscript fully available?

Reviewer #3: Yes

5. Is the manuscript presented in an intelligible fashion and written in standard English?

Reviewer #3: Yes

6. Review Comments to the Author

Reviewer #3: After careful revision, the authors have addressed all my concerns. The manuscript can be accepted.

7. PLOS authors have the option to publish the peer review history of their article (what does this mean?). If published, this will include your full peer review and any attached files.

Reviewer #3: No

---

## [Author Response · Author response to Decision Letter 3]

28 Oct 2024

The entire manuscript has been thoroughly reviewed, with a particular emphasis on figure 2. The figure was previously blurry and unreadable due to its size, but I have since replaced it with the actual figure. I have attempted to modify the figure with the assistance of PACE, but it remains blurry. Consequently, the original figure has been incorporated into the manuscript. Additionally, the final paragraph of heading 4.2 has been updated. I am grateful for the comments and suggestions of all the reviewers, as they have been extremely helpful in the process of modification. Thank you again for your insightful remarks and recommendations.

---

## [Editor Report · Decision Letter 4]

22 Dec 2024

Using the TSA-LSTM Two-Stage Model to Predict Cancer Incidence and Mortality

PONE-D-23-37813R4

Dear Dr. Khan 

We’re pleased to inform you that your manuscript has been judged scientifically suitable for publication and will be formally accepted for publication once it meets all outstanding technical requirements.

Kind regards,

Amgad Muneer

Academic Editor

PLOS ONE
---

## [Editor Report · Acceptance letter]

3 Jan 2025

PONE-D-23-37813R4 

PLOS ONE

Dear Dr. Khan, 

I'm pleased to inform you that your manuscript has been deemed suitable for publication in PLOS ONE. Congratulations! Your manuscript is now being handed over to our production team.

Kind regards, 

on behalf of

Dr. Amgad Muneer 

Academic Editor

PLOS ONE